# Development of intensiometric indicators for visualizing N-cadherin interaction across cells

Takashi Kanadome [1,2], Kanehiro Hayashi[3], Yusuke Seto[4], Mototsugu Eiraku[4,5], Kazunori Nakajima [3], Takeharu Nagai [2] & Tomoki Matsuda [2✉]

N-cadherin (NCad) is a classical cadherin that mediates cell–cell interactions in a $Ca^{2+}$-dependent manner. NCad participates in various biological processes, from ontogenesis to higher brain functions, though the visualization of NCad interactions in living cells remains limited. Here, we present intensiometric NCad interaction indicators, named INCIDERs, that utilize dimerization-dependent fluorescent proteins. INCIDERs successfully visualize reversible NCad interactions across cells. Compared to FRET-based indicators, INCIDERs have a ~70-fold higher signal contrast, enabling clear identification of NCad interactions. In primary neuronal cells, NCad interactions are visualized between closely apposed processes. Furthermore, visualization of NCad interaction at cell adhesion sites in dense cell populations is achieved by two-photon microscopy. INCIDERs are useful tools in the spatiotemporal investigation of NCad interactions across cells; future research should evaluate the potential of INCIDERs in mapping complex three-dimensional architectures in multi-cellular systems.

[1] Precursory Research for Embryonic Science and Technology (PRESTO), Japan Science and Technology Agency (JST), Kawaguchi, Saitama 332-0012, Japan. [2] Department of Biomolecular Science and Engineering, SANKEN (The Institute of Scientific and Industrial Research), Osaka University, 8-1 Mihogaoka, Ibaraki 567-0047, Japan. [3] Department of Anatomy, Keio University School of Medicine, Shinjuku-ku, Tokyo 160-8582, Japan. [4] Laboratory of Developmental Systems, Institute for Life and Medical Sciences, Kyoto University, Kyoto 606-8507, Japan. [5] Institute for the Advanced Study of Human Biology (WPI-ASHBi), Kyoto University, Kyoto 606-8507, Japan. ✉email: tmatsuda@sanken.osaka-u.ac.jp

In multicellular organisms, cell adhesion plays an essential role in cell sorting[1–3], the formation and maintenance of tissues and organs[4,5], and synaptic transmission[6–8]. In cell adhesion, the adhesive molecules govern the interactions across the cells. Neural cadherin (N-cadherin, hereafter NCad) a cell adhesion molecule that belongs to the classical cadherin family and mediates $Ca^{2+}$-dependent cell adhesions[9]. NCad contributes to a broad range of biological processes, including ontogenesis[10], cell migration[11–13], neurite outgrowth[14,15], and synaptic potentiation[16,17]; though, due to a lack of appropriate detection techniques, it remains unclear when and where NCad interactions occur. To date, only one NCad interaction indicator has been developed[18]. The indicator exploits a Förster resonance energy transfer (FRET) mechanism between two fluorescent proteins, which is widely used in the development of fluorescent indicators of various cellular events[19–22]. Using the FRET-based NCad interaction indicator, Kim et al.[18] investigated the $Ca^{2+}$-responsive dynamics of NCad interactions across cells. However, FRET-based indicators sometimes encounter a hurdle to their broad application due to their low signal contrast that causes difficulty in detecting FRET signals.

Intensiometric indicators are considered effective alternatives for detecting protein conformation changes and protein–protein interactions via an alteration in fluorescence intensity. For the visualization of cell–cell interactions, split-GFP technique can be used. GRASP[23], mGRASP[24], SynView[25], and eGRASP[26] have been developed based on functional complementation between split superfolder GFP fragments (named GFP1-10 and GFP11)[27]. Applications in brain studies have achieved partial visualization of connectomes with a high signal contrast. A recently developed GRAPHIC system, utilizing the fragments GFP1-7 and GFP8-11, successfully visualized cell–cell interactions and neuronal connections with comparatively high signal intensity[28,29]. However, visualization of cell–cell interactions using this technology is inhibited by some limitations. For example, the emergence of fluorescence from the reconstituted split components by cell–cell interactions requires a time delay derived from the maturation of chromophores, and the irreversibility of the process makes it impossible to monitor the dissociation of cell–cell interactions[25,28,30].

As a technique to visualize protein–protein interactions, a dimerization-dependent fluorescent protein (ddFP) has recently been adopted for various genetically encoded indicators to monitor apoptosis[31], $Ca^{2+}$ levels[31], kinase activity[31], phosphatidylinositol 4,5-bisphosphate ($PIP_2$) levels[32], and small GTPase activity[33]. A ddFP consists of two fluorescent protein-monomers, namely ddFP-A and ddFP-B that has and lacks chromophores, respectively. The chromophore in ddFP-A, which is quenched in the monomeric state, emits bright fluorescence when forming a heterodimer with ddFP-B. ddFP was originally developed as a red variant ddRFP from dTomato[34] and followed by a green variant ddGFP and a yellow variant ddYFP[35]. These color variants give applicability on multicolor imaging and combination with optogenetic tools[33]. Unlike split-GFP technique, ddFPs do not require a time delay after protein–protein interactions and are reversible, which allows for real-time monitoring of transitions between interactions[31–35].

In this study, we evaluate the efficacy and practicality of INCIDERs: fluorescent indicators for the detection of NCad interactions. INCIDERs with reversible properties overcome the limitations of split-GFP-based indicators and were successfully applied in the monitoring of the association and dissociation of NCad interactions in living cells. We successfully visualized NCad interactions in primary neurons and spheroids consisting of multi-layered cells.

## Results

**Design of ddFP-based NCad interaction indicators (INCIDERs).** To generate intensiometric NCad interaction indicators,

we referred to the FRET-based indicator for NCad interaction[18]. This indicator contains a cyan fluorescent protein (Cerulean) and a yellow fluorescent protein (Venus) as the FRET donor and acceptor, respectively; each fluorescent protein is inserted into the respective NCad molecules. Their insertion site (the 311th residue) at a second extracellular cadherin (EC) domain was near the interface of intercellular NCad interaction, and the fluorescent protein insertion did not impair the localization or function of NCad[18]. By replacing the fluorescent proteins with ddGFP-A and ddFP-B, respectively, we generated ddGFP-A-inserted NCad (NCad-GA) and ddFP-B-inserted NCad (NCad-B) (Fig. 1a). We used two variants of ddFP-B with different affinities to ddGFP-A, ddFP-B1 and ddFP-B3 ($K_d = 3$ μM and 40 μM, respectively)[31]. The indicators composed of the combinations of NCad-GA/NCad-B1 and NCad-GA/NCad-B3 were denominated as INCIDER (Indicator for N-Cadherin Interaction upon DimERization)-1 and INCIDER-3, respectively. To evaluate heterodimer formation and the associated fluorescence, we transiently co-expressed the INCIDERs in HEK293T cells. While some artificial fluorescent puncta, which are considered autofluorescence were observed in the cells individually expressing components of the INCIDER (NCad-GA, NCad-B1, or NCad-B3), there was no fluorescence at their cell–cell interfaces. On the other hand, cells co-expressing INCIDER-1 and INCIDER-3 fluoresced at the cell adhesion site and intracellular space, as did the fluorescence localization of Venus-inserted NCad (NCad-V), indicating heterodimer formation of INCIDERs in intra- and intercellular spaces (Supplementary Fig. 1).

**Detection of intercellular NCad interaction by INCIDERs.** Since INCIDERs are based on the co-expression of two components in the same cell, intracellular and intercellular NCad interactions become indistinguishable. To specifically monitor intercellular NCad interactions, we co-cultured cells expressing only one of the INCIDER components, i.e., NCad-GA or NCad-B. Cells expressing NCad-GA and NCad-B were distinguished by the nuclear localized fluorescent marker proteins mCherry and miRFP670, respectively. Fluorescence was observed at the cell adhesion sites between mCherry-positive and miRFP670-positive cells (Fig. 1b), which verified the detection of intercellular NCad interactions by INCIDERs. In a comparison of fluorescence intensity between INCIDER-1 and INCIDER-3, INCIDER-1 had a slightly higher signal (Fig. 1c), which was expected given the higher heterodimer affinity between ddGFP-A and ddFP-B1. INCIDER fluorescence signals were positively correlated with their expression levels, as estimated by their expression marker fluorescent proteins (Supplementary Fig. 2a, b).

It is possible that the heterodimerization between ddGFP-A and ddFP-B enhances association of cell–cell interactions. To address the possibility, we examined the adhesive functions of INCIDERs using K562 cells that lack endogenous adhesion proteins[36]. While K562 cells that were not transfected with exogeneous NCad constructs were observed as separated cells, those expressing exogeneous NCad constructs formed cell aggregates (Supplementary Fig. 3a). Compared with cells expressing non-tagged NCad, the size of cell aggregates expressing ddFP-based NCad (INCIDER) and FRET-based NCad indicators (Ncad-FRET) tended to be smaller. However, INCIDER and Ncad-FRET induced the formation of cell aggregates to the same extent (Supplementary Fig. 3b). This result shows that the insertion of fluorescent proteins into NCad slightly weakens the adhesive function, but heterodimerization of ddGFP-A and ddFP-B does not significantly enhance association of cell–cell interactions.

To examine the importance of the ddGFP-insertion site for the detection of NCad interactions, we generated NCad with ddGFP

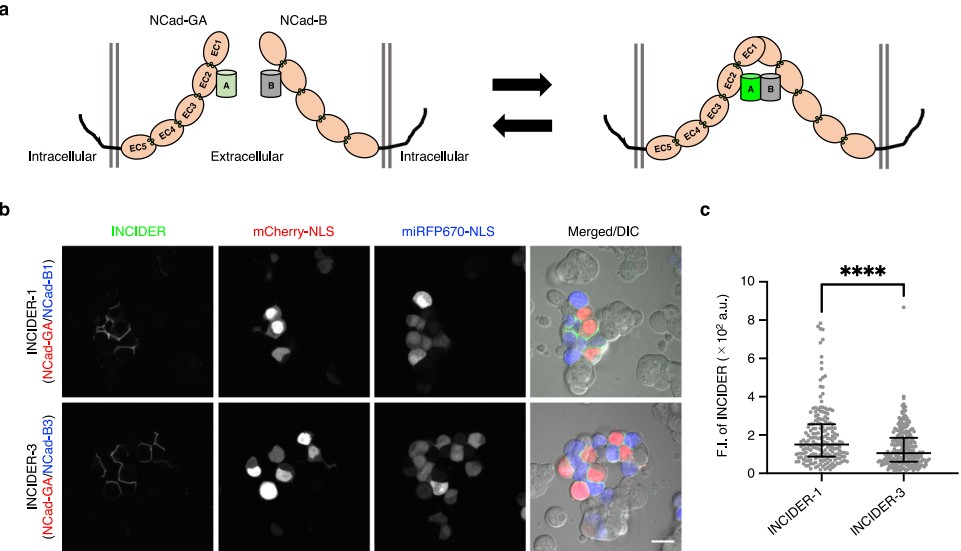

**Fig. 1 Design of an intensiometric NCad interaction indicator using ddGFP. a** Schematics of intercellular NCad interactions by INCIDERs. NCad is represented by five serially repeated extracellular domains (EC domains). In NCad-GA and NCad-B constructs, ddGFP-A and ddFP-B are inserted in the EC2 domain of NCad. By binding of NCad-GA with NCad-B, ddGFP-A and ddFP-B form a heterodimer, resulting in green fluorescence. **b** Fluorescence and DIC images of intercellular NCad interaction. HEK293T cells individually expressing INCIDER components, i.e. NCad-GA and NCad-B (NCad-B1 or NCad-B3) were co-cultured, and then observed using a confocal microscope. mCherry-NLS and miRFP670-NLS were bicistronically co-expressed with NCad-GA and NCad-B, respectively to distinguish cells. Scale bar, 20 μm. **c** INCIDER fluorescence signals at the cell adhesion sites were quantified and compared between INCIDER-1 and INCIDER-3. Results are presented as lower quartile (lower whisker), median (center line), and upper quartile (upper whisker). A significant difference was analyzed by Mann–Whitney U test. ****p < 0.0001. 189 (INCIDER-1) and 244 (INCIDER-3) cell adhesion sites were analyzed from two independent experiments.

inserted at a site distant to the intercellular NCad interaction interface, referred to as NCad-GA$_{prox}$, NCad-B1$_{prox}$, and NCad-B3$_{prox}$. (Supplementary Fig. 4a). As in the previous report, NCad in which Venus was inserted at the same site (NCad-V$_{prox}$) localized at the cell adhesion sites[18], indicating that the insertion of a fluorescent protein at this site does not impair NCad localization (Supplementary Fig. 4b). Fluorescence of HEK293T-cells co-expressing NCad-GA$_{prox}$ and NCad-B$_{prox}$ (NCad-B1$_{prox}$ or NCad-B3$_{prox}$) indicated heterodimer formation between ddGFP-A and ddFP-B predominantly in the intracellular space (Supplementary Fig. 4b). Cells individually expressing NCad-GA$_{prox}$ or NCad-B$_{prox}$ were co-cultured to evaluate the efficacy of NCad-GA$_{prox}$/NCad-B$_{prox}$ for visualization of intercellular NCad interactions. Compared to INCIDERs, little fluorescence was observed at the cell adhesion sites between NCad-GA$_{prox}$-expressing cells and NCad-B$_{prox}$-expressing cells (Supplementary Fig. 4c). These results demonstrate the importance of the insertion site for heterodimerization of ddGFP-A and ddFP-B across cells.

**Visualization of intercellular NCad interaction by co-expressed INCIDER components**. INCIDERs are supposed to form not only a *trans*-interaction across the opposed plasma membranes of different cells, but also a *cis*-interaction on the individual membrane[37,38]. Cells co-expressing INCIDER components fluoresced at cell adhesion sites (Supplementary Fig. 1). This prompted us to examine, in a co-culture experiment, how *trans*- and *cis*-interactions contribute to the fluorescence signal (Fig. 2a). We prepared two groups of cells: (i) cells co-expressing NCad, lacking a fluorescent protein, with an expression marker EBFP2-NLS and (ii) cells co-expressing INCIDER components (NCad-GA and NCad-B) with an expression marker mCherry-NLS. If

fluorescence is detected between EBFP2-positive and mCherry-positive cells (B-R), it indicates that INCIDERs fluoresce with *cis*-interactions. On the other hand, if fluorescence is detected only between mCherry-positive cells (R-R), the fluorescence of INCIDERs would be solely derived from *trans*-interactions. We observed fluorescence of INCIDERs at R-R (Fig. 2b, open arrowheads), but not at B-R (Fig. 2b, closed arrowheads). While NCad-V showed low contrast of Venus signals between R-R and B-R (1.37), those of INCIDER-1 and -3 signals between R-R and B-R were high (5.31 and 7.72, respectively) (Fig. 2c). These results indicate that co-expressed INCIDER components predominantly visualize *trans*-interactions.

**Confirmation of signal specificity for NCad interactions**. Since ddGFP-A and ddFP-B have an affinity for heterodimer formation, INCIDERs might generate signals only via the self-interaction of ddGFP-A and ddFP-B regardless of NCad interactions. To address this issue, we evaluated the activity of a mutant NCad that can form neither *trans*- nor *cis*-interactions. We introduced W2A/R14E and V81D/V174D mutations to eliminate *trans*- and *cis*-interactions, respectively[37–40]. To exclude the effect of endogenous NCad, we used L cells that lack endogenous cadherins[41] and established stable cell lines of fluorescent protein-inserted wild-type NCad (NCad$^{WT}$-V, NCad$^{WT}$-GA, NCad$^{WT}$-B1, and NCad$^{WT}$-B3) and fluorescent protein-inserted mutant NCad (NCad$^{mut}$-V, NCad$^{mut}$-GA, NCad$^{mut}$-B1, and NCad$^{mut}$-B3). We confirmed, by western blot analysis using an anti-NCad antibody, that NCad was detected as two bands (upper and lower bands corresponding to unprocessed and processed NCad, respectively) and expression levels of total NCad varied between cell lines to some extent (Fig. 3a, b). In the stable cell line, NCad$^{mut}$-V was localized at the plasma membrane and cell–cell interface, similar

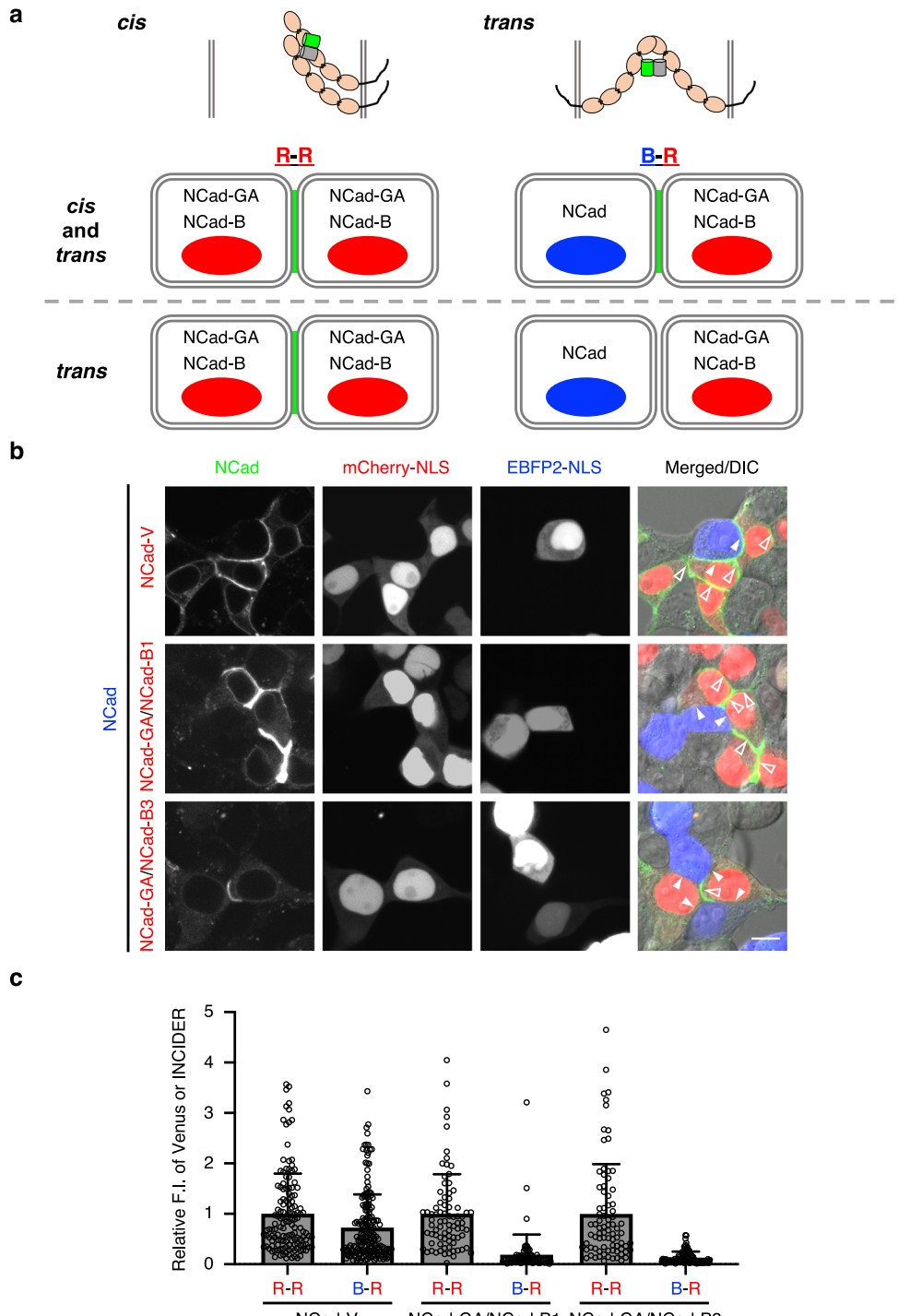

**Fig. 2 Trans-interface visualized by INCIDERs. a** Schematic representation of the co-culture experiment between cells co-expressing INCIDER components (NCad-GA/NCad-B) and cells expressing NCad without a fluorescent protein. **b** HEK293T cells expressing the indicated constructs were co-cultured and observed using a confocal microscope. mCherry-NLS and EBFP2-NLS are hallmarks of NCad-V or NCad-GA/NCad-B and NCad without fluorescent protein-expressing cells, respectively. Cell adhesion sites between red cells and between blue cells and red cells are described as R-R and B-R, respectively. Closed and open arrowheads indicate cell adhesion sites between mCherry-positive cells/EBFP2-positive cells (B-R) and mCherry-positive cells (R-R), respectively. Scale bar, 10 μm. **c** Fluorescence intensities of Venus and INCIDERs at R-R and B-R cell adhesion sites were quantified and normalized by the mean values of R-R cell adhesion sites. Data are presented as relative mean values ± SD. 137 (R-R of NCad-V), 184 (B-R of NCad-V), 80 (R-R of INCIDER-1), 77 (B-R of INCIDER-1), 81 (R-R of INCIDER-3), and 93 (B-R of INCIDER-3) cell adhesion sites were analyzed from three independent experiments.

to NCad$^{WT}$-V (Fig. 3c). We co-cultured L cells stably expressing NCad$^{mut}$-GA and NCad$^{mut}$-B, respectively. Although ddGFP signals were detected at the cell adhesion sites between NCad$^{WT}$-GA-expressing and NCad$^{WT}$-B-expressing cells, significantly

fewer signals were detected at the cell–cell interfaces between NCad$^{mut}$-GA-expressing and NCad$^{mut}$-B-expressing cells (Fig. 3c, d). It is possible that a difference in membrane localization efficiency between NCad$^{WT}$ and NCad$^{mut}$ contributed to a difference

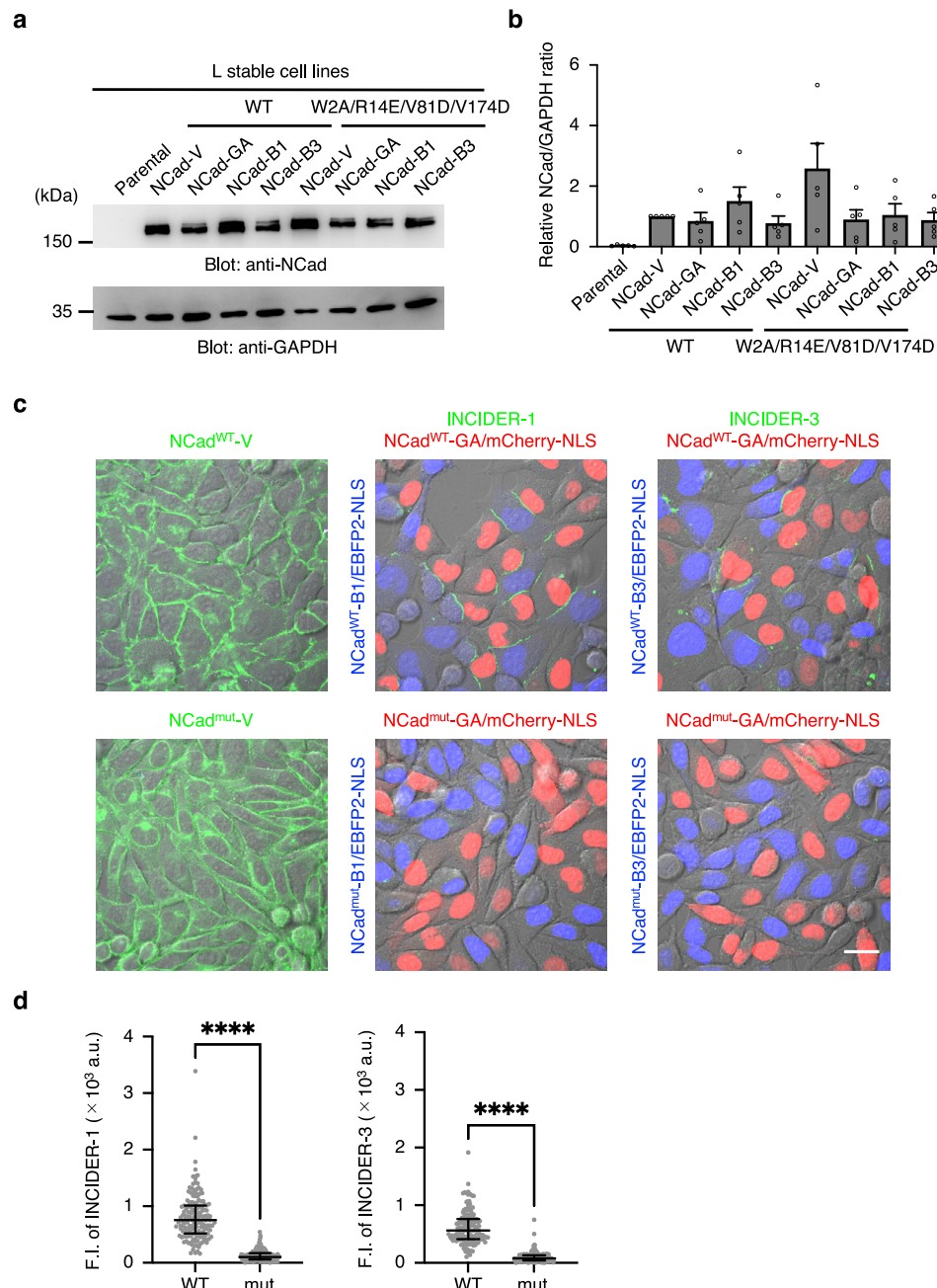

**Fig. 3 NCad-mediated cell–cell interactions visualized by INCIDERs. a** Examination of NCad expression in stable cell lines. Lysates prepared from parental L cells and L cells stably expressing the indicated NCad constructs were subjected to SDS-PAGE, followed by western blot analysis using the indicated antibodies. **b** Relative protein expression levels were estimated among cell lines as NCad/GAPDH ratio based on western blot analysis data. Both upper and lower bands of NCad were used for the quantification. NCad/GAPDH ratios were normalized by that of NCad$^{WT}$-V. Data are presented as relative mean values ± SEM. Five independent experiments were performed. **c** The localization of NCad$^{WT}$-V and NCad$^{mut}$-V in stable cell lines. Co-culture experiments of cells individually expressing the NCad$^{WT}$-GA/NCad$^{WT}$-B1 pair (INCIDER-1), NCad$^{WT}$-GA/NCad$^{WT}$-B3 pair (INCIDER-3), NCad$^{mut}$-GA/NCad$^{mut}$-B1 pair, and NCad$^{mut}$-GA/NCad$^{mut}$-B3 pair. Scale bar, 20 µm. **d** Wild-type (WT) and mutant (mut) INCIDERs' fluorescent signals at cell–cell contact sites were quantified and compared. Data are presented as lower quartile (lower whisker), median (center line), and upper quartile (upper whisker). Significant differences were analyzed by Mann–Whitney $U$ test. ****$p < 0.0001$. 166 (WT of INCIDER-1), 156 (mut of INCIDER-1), 139 (WT of INCIDER-3), and 227 (mut of INCIDER-3) cell–cell contact sites from three independent experiments were analyzed.

in ddGFP signals. To address this possibility, we performed surface staining of NCad$^{WT}$-V-expressing cells and NCad$^{mut}$-V-expressing cells and estimated the efficiency by calculating the ratio of immunoreactive signals using anti-GFP antibody to Venus signals. We found that NCad$^{mut}$-V localized at the plasma membrane with approximately 60% efficiency of NCad$^{WT}$-V (Supplementary Fig. 5a, b). This result implies that poorer ddGFP

signals from NCad$^{mut}$-GA and NCad$^{mut}$-B were partially contributed to by poorer membrane localization. Since a difference in membrane localization was not so high compared with ddGFP signals, we concluded that INCIDERs give fluorescence dependent on the NCad-mediated cell–cell interaction.

We further performed the co-culture in the combination of NCad$^{WT}$/NCad$^{mut}$. Both NCad$^{WT}$-GA/NCad$^{mut}$-B1 pair and

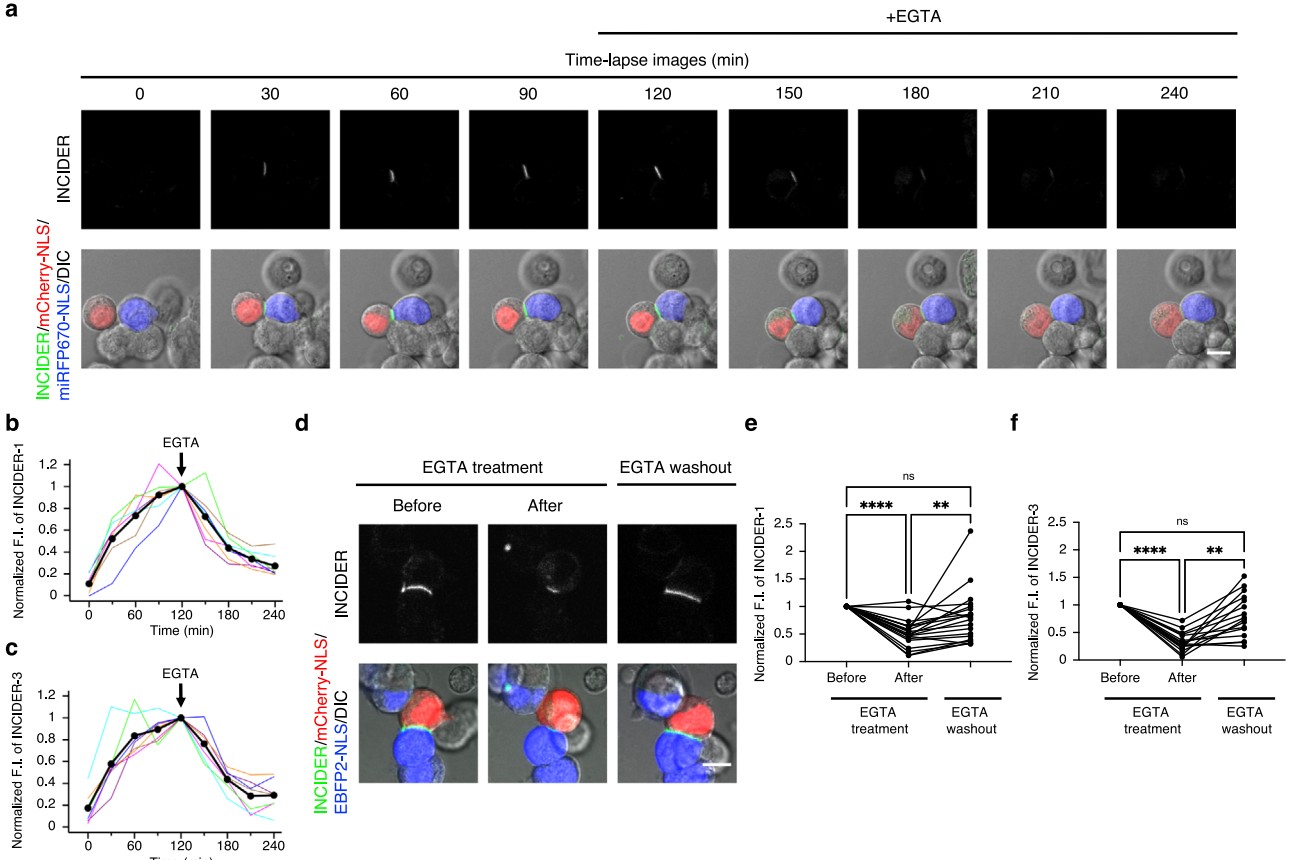

**Fig. 4 Monitoring of the dynamics of NCad interactions. a** Time-lapse images of INCIDER-1. HEK293T cells individually expressing INCIDER-1 components (NCad-GA and NCad-B1) were observed for 120 min immediately after seeding. 10 mM EGTA was administrated 120 min after seeding, and cells were imaged for a further 120 min. NCad-GA and NCad-B1 expressing cells were marked by mCherry-NLS and miRFP670-NLS, respectively. Scale bar, 10 μm. **b**, **c** The normalized fluorescence intensities at the cell adhesion sites formed by INCIDER-1 (**b**) and INCIDER-3 (**c**) were plotted over time. Values were normalized by the fluorescence intensity of INCIDER at 120 min. Colored lines represent the fluorescence changes from individual cell adhesion sites, and black circles indicate the averaged time course. 7 cell adhesion sites (both INCIDER-1 and INCIDER-3) from seven (INCIDER-1) and six (INCIDER-3) independent experiments were analyzed. Significant differences at each time point between INCIDER-1 and INCIDER-3 were not confirmed by a Mann–Whitney $U$ test ($p > 0.5$ in all time points). **d** The effect of EGTA treatment and washout on the fluorescence intensity of INCIDER-1. HEK293T cells individually expressing INCIDER-1 components (NCad-GA and NCad-B1) were observed at 1 h after seeding. After 2 h from 10 mM EGTA treatment, cells were washed to remove EGTA and further cultured for 3 h under a microscope. Scale bar, 10 μm. **e**, **f** Changes in fluorescence intensities of INCIDER-1 (**e**) and INCIDER-3 (**f**) at the cell adhesion sites were quantified. Values were normalized by the fluorescence intensity of INCIDER before EGTA treatment. Significant differences were analyzed by Friedman test, followed by Dunn's multiple comparison test. ****$p < 0.0001$, **$p < 0.01$, ns indicates $p > 0.05$. 20 (INCIDER-1) and 16 (INCIDER-3) cell adhesion sites from eight (INCIDER-1) and ten (INCIDER-3) independent experiments were analyzed.

NCad$^{mut}$-GA/NCad$^{WT}$-B1 pair showed ddGFP signals that were lower than the NCad$^{WT}$-GA/NCad$^{WT}$-B1 pair (Supplementary Fig. 5c, d). This result raises a possibility that NCad$^{WT}$ weakly interacts with NCad$^{mut}$. However, further analysis is required to reach a conclusion as we did not examine the interaction between NCad$^{WT}$ and NCad$^{mut}$.

**Time-lapse imaging of reversible NCad interaction**. To assess the detectability of reversible intercellular NCad interactions, we performed time-lapse imaging immediately after seeding cells expressing individual INCIDER components. During the formation of cell–cell contact, the INCIDER fluorescence signal increased at the cell adhesion sites. After the addition of EGTA to the culture medium (for the chelation of extracellular $Ca^{2+}$ that is essential for intercellular NCad interaction[9]), the INCIDER signal gradually decreased over time (Fig. 4a). INCIDER-1 and INCIDER-3 showed similar time trajectories related to fluorescence intensity and there was no significant difference between them, regardless of the differences in affinity to ddGFP-A. Signal

intensity reached 50% ~50 min after the addition of EGTA (Fig. 4b, c). We further examined whether INCIDERs can monitor a re-association of NCad. INCIDER signals that declined with EGTA treatment were increased with the washout of EGTA again (Fig. 4d–f). These results indicate that INCIDERs show the appropriate availability for monitoring the transition between association and dissociation of NCad interactions.

**Comparison between INCIDERs and conventional indicators for cell–cell interactions**. To determine relative performance, we compared the signal contrast between FRET-based NCad indicator (Ncad-FRET) and INCIDERs. We individually expressed and subsequently co-cultured Cerulean-inserted NCad (C-Ncad) and Venus-inserted NCad (V-Ncad) in COS7 cells. FRET was confirmed at the cell adhesion sites as a YFP/CFP ratio (Fig. 5a). We measured the YFP/CFP ratio at the cell adhesion sites and non-cell adhesion surfaces of cells expressing C-Ncad. The signal contrast for FRET was calculated by dividing the YFP/CFP ratio at cell adhesion sites by that at non-cell adhesion sites (Fig. 5c).

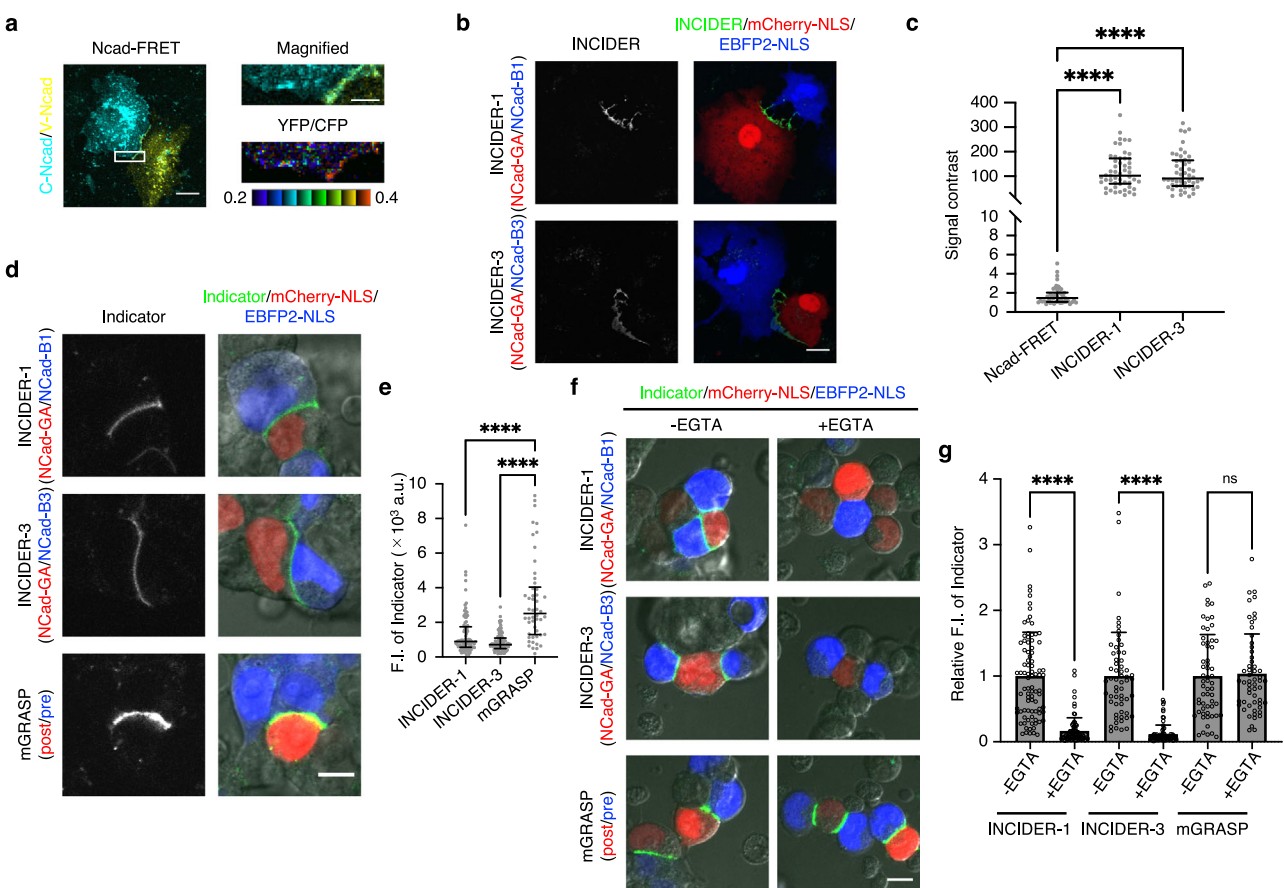

**Fig. 5 Comparison between INCIDERs and conventional indicators for cell–cell interactions. a** NCad interaction monitored by the FRET-based NCad indicator (Ncad-FRET) in COS7 cells. COS7 cells expressing the indicated constructs were co-cultured. The FRET signal at the cell adhesion site is displayed as a YFP/CFP ratio (bottom right). Scale bar, 20 μm (left), 5 μm (upper right). **b** NCad interaction monitored by INCIDERs in COS7 cells. COS7 cells expressing the indicated constructs were co-cultured. mCherry-NLS and EBFP2-NLS are hallmarks of NCad-GA and NCad-B expressing cells, respectively. Scale bar, 20 μm. **c** The signal contrast between cell adhesion sites and non-cell adhesion sites. The signal contrast of Ncad-FRET was calculated by dividing the YFP/CFP ratio at the cell adhesion sites by that at the non-cell adhesion sites of the cells expressing C-Ncad. The signal contrast of INCIDERs was calculated dividing the fluorescence intensity of INCIDERs at the cell adhesion sites by that at the non-cell adhesion sites of the cells expressing NCad-GA. Data are presented as lower quartile (lower whisker), median (center line), and upper quartile (upper whisker). A significant difference was shown by a Kruskal-Wallis test, followed by a Dunn's multiple comparison test. ****$p < 0.0001$. 50 (Ncad-FRET), 52 (INCIDER-1), and 50 (INCIDER-3) cell adhesion sites from four independent experiments were analyzed. **d** Comparison of brightness between INCIDERs and mGRASP. HEK293T cells transiently expressing the indicated components were co-cultured. Scale bar, 10 μm. **e** Fluorescence intensities of the indicators at the cell adhesion sites were quantified. Data are presented as lower quartile (lower whisker), median (center line), and upper quartile (upper whisker). Significant differences were analyzed by Kruskal-Wallis test, followed by Dunn's multiple comparison test. ****$p < 0.0001$. 90 (INCIDER-1), 98 (INCIDER-3), and 50 (mGRASP) cell adhesion sites from two independent experiments were analyzed. **f** Comparison of reversibility between INCIDERs and mGRASP. HEK293T cells transiently expressing the indicated components were co-cultured for 6 h, and then co-cultured in the presence or absence of 10 mM EGTA for further 2 h. Scale bar, 10 μm. **g** Fluorescence intensities of the indicators at the cell–cell contact sites in the presence or absence of 10 mM EGTA were quantified and normalized by the mean values in the absence of 10 mM EGTA. Data are presented as relative mean values ± SD. Significant differences were analyzed by Mann–Whitney U-test. ****$p < 0.0001$. 87 (-EGTA, INCIDER-1), 81 (+EGTA, INCIDER-1), 59 (-EGTA, INCIDER-3), 76 (+EGTA, INCIDER-3), 59 (-EGTA, mGRASP), 56 (+EGTA, mGRASP) cell–cell contact sites from three independent experiments were analyzed.

INCIDERs also visualized NCad interactions at the COS7 cell adhesion sites (Fig. 5b). The signal contrast for INCIDERs was calculated by dividing the fluorescence intensity at the cell adhesion sites by that at the non-cell adhesion sites of NCad-GA expressing cells. The signal contrast of INCIDERs was ~70-times higher than that of Ncad-FRET (Fig. 5c). INCIDERs enabled the clear identification of NCad interactions across cells without post-hoc image processing.

Next, we compared the performance between INCIDERs and a split-GFP-based indicator for cell–cell interactions (mGRASP[24]). mGRASP showed green fluorescence at the cell adhesion sites similar to INCIDERs and its signal was more than twice as high

as that of INCIDERs (Fig. 5d, e). Split-GFP-based indicators for cell–cell interactions are potentially unable to monitor the dissociation of cell–cell interactions[25,28,30]. To evaluate the relative dissociation property, we examined the effect of EGTA treatment on pre-formed cell–cell interactions. We first co-cultured HEK293T cells individually expressing INCIDERs or mGRASP components for 6 h and then further cultured them in the presence or absence of 10 mM EGTA for 2 h. The INCIDER signals were diminished by incubation with EGTA while mGRASP showed signals regardless of EGTA treatment (Fig. 5f, g). This result shows that INCIDERs can monitor the dissociation of cell–cell interactions unlike split-GFP-based indicators.

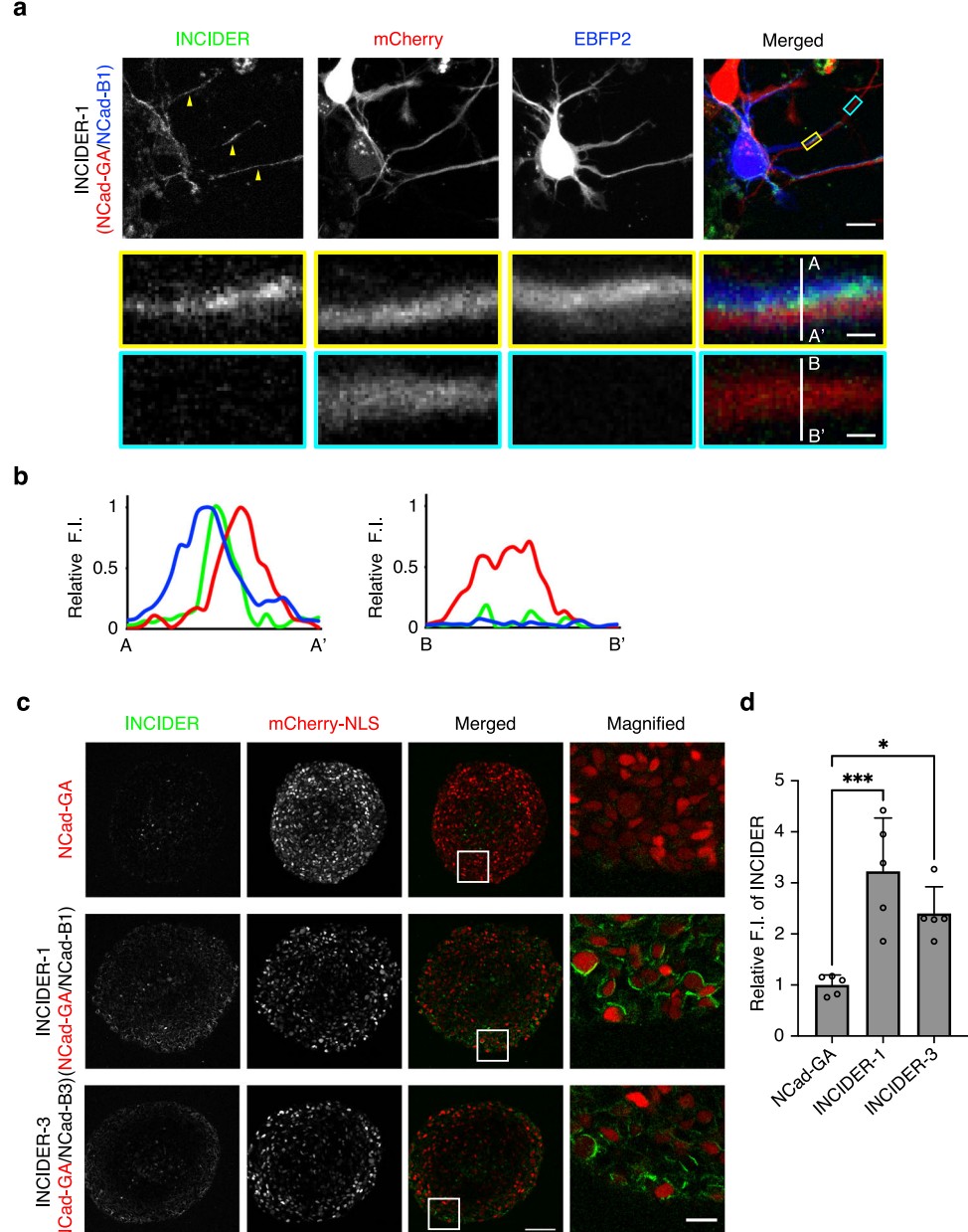

**Fig. 6 Application of INCIDERs to various samples. a** Fluorescence images of intercellular NCad interaction in neurons visualized by INCIDER-1. Dissociated cortical neurons individually expressing INCIDER-1 components (NCad-GA and NCad-B1) were co-cultured and observed using a confocal microscope at 2 DIV. mCherry and EBFP2 were bicistronically co-expressed with NCad-GA and NCad-B1, respectively, to differentiate cells and visualize cell morphology. Scale bar, 20 μm (upper) and 5 μm (lower). 102 fields of view from six independent experiments were observed. **b** Fluorescence intensities along the white lines across a neuronal process of an NCad-GA-expressing neuron with (from A to A') or without (from B to B') a process of an NCad-B1-expressing neuron. Relative fluorescence intensities of mCherry, EBFP2, and INCIDER are represented by red, blue, and green, respectively. **c** Intercellular NCad interaction in spheroids visualized by INCIDERs. Spheroids formed by the L cells stably expressing the indicated NCad constructs were observed using a two-photon microscope. mCherry-NLS is a hallmark of NCad-GA-expressing cells. Scale bar, 100 μm (left) and 25 μm (right). **d** Mean values of green fluorescence from the indicated spheroids were quantified and normalized by that of the spheroid consisting of NCad-GA-expressing cells. Data are presented as relative mean values ± SD. Significant differences were analyzed by ordinary one-way ANOVA, followed by Dunnett's multiple comparisons test, with a single pooled variance. ***$p < 0.001$, *$p < 0.05$. In all samples, 5 spheroids from two independent experiments were analyzed.

**Efficacy of INCIDER visualization of NCad interactions in different specimens.** Finally, we assessed the performance and application potential of INCIDERs across a range of samples. NCad interaction in immature neurons is involved in the determination of neuronal polarity[15] and neurite growth and maintenance[14]. To examine the detectability of INCIDERs in neurons, we expressed individual INCIDER components in

primary cultured cortical neurons and assessed fluorescence after 2 days in vitro (DIV). As expected, the INCIDER fluorescence signal was predominantly observed at the contact sites of neuronal processes (Fig. 6a, Supplementary Fig. 6). The signal was specifically detected between closely apposed processes of an NCad-GA-expressing neuron and an NCad-B-expressing neuron (Fig. 6b). By contrast, a process of an NCad-GA-expressing

neuron that was not attached to an NCad-B-expressing neuron showed little fluorescence (Fig. 6b). We also applied Ncad-FRET to neurons. However, FRET signal was not detected well at the contact sites between C-Ncad-expressing neurons and V-Ncad-expressing neurons, which was possibly due to the low signal contrast of Ncad-FRET. (Supplementary Fig. 7).

We examined the functionality of INCIDERs in multi-layered spheroids expressing INCIDER components. Although NCad stable cell lines formed spheroids, the parental L cells failed to form spheroids (Supplementary Fig. 8). Using a two-photon microscope, we confirmed that INCIDER fluorescence signals were detected in spheroids formed by NCad-GA-expressing cells and NCad-B-expressing cells (Fig. 6c, d). These results suggest that INCIDERs can be applied to a range of specimens with different physical characteristics.

## Discussion

Here, we developed ddFP-based indicators for NCad interactions by inserting ddFPs in NCad. We showed that INCIDER signals were dependent on NCad interactions, not the self-interaction of ddGFP-A and ddFP-B (Fig. 3). However, INCIDER should generate the signal via the ddGFP-A/ddFP-B heterodimerization. This notion raises an interpretation that NCad interacting property of INCIDER leads to the formation of a ddGFP-A/ddFP-B heterodimer. Therefore, this ddGFP-A/ddFP-B heterodimerization possibly affects cell–cell interactions. Cell aggregation assay using K562 cells showed that INCIDERs induced cell aggregation to the same extent as Ncad-FRET, indicating that the heterodimerization of ddGFP-A/ddFP-B does not significantly induce cell–cell interactions (Supplementary Fig. 3). However, EGTA treatment experiments showed that time scale of fluorescence decay of INCIDERs upon EGTA treatment was in the order of minute-to-hour (Fig. 4a–c) although that of Ncad-FRET has been reported to be in the order of seconds[18]. This means that the heterodimerization of ddGFP-A/ddFP-B possibly affects the dissociation of NCad interactions. While INCIDERs are reversible unlike split-GFP-based indicators (Fig. 5f, g), it is noticeable that the relatively slow decay is a limitation of INCIDERs. The development of ddFP-B mutants with lower affinity to ddGFP-A may solve this limitation.

We used ddFP-B1 and ddFP-B3, respectively, to prepare ddFP-B-inserted NCad (NCad-B1 and NCad-B3). The $K_d$ of ddGFP-A to ddFP-B1 and ddFP-B3 are 3 μM and 40 μM, respectively[31]. Since the $K_d$ of the homodimerization of NCad is approximately 25 μM[42], NCad-B3 would be less likely to interfere with NCad interactions. However, there was no significant difference in time-dependent fluorescence intensity between INCIDER-1 and INCIDER-3 (Fig. 4b, c). Fluorescence intensity was significantly higher in INCIDER-1 compared to INCIDER-3 (Fig. 1c). Thus, INCIDER-1 seems to be the most suitable indicator for monitoring NCad interactions, though further experimental comparisons between these INCIDERs are required to comprehensively determine their respective advantages.

INCIDERs function not only as indicators, but also adhesion molecules that promote cell–cell interactions by maintaining intercellular distance. This property can disturb endogenous NCad functions or confer a gain-of-function by overexpression of interactions. To promote the accuracy of assays, the expression level should be adjusted by, for example, establishing stable cell lines or knock-in cells. The co-expression of INCIDER components (NCad-GA and NCad-B) predominantly visualized a *trans*-interface of NCad interactions at the cell–cell junctions (Fig. 2), independent of whether two adjacent cells express different components of INCIDER. Therefore, the effective monitoring of NCad interactions could be achieved by using knock-in

cells or knock-in mice expressing NCad-GA and NCad-B heterozygously.

Split-GFP-based indicators for cell–cell interactions, such as m- or eGRASP and SynView, have aided our understanding of connectomes by direct visualization of neural connections. However, their irreversibility does not allow for real-time monitoring of connectome changes, which the ddFP-based technique in the current study could achieve with an appropriate level of fluorescence.

In summary, we developed intensiometric NCad interaction indicators, INCIDERs, utilizing ddGFP that can reversibly visualize NCad interactions across cells. The INCIDERs can be applied to a wide range of specimens and could provide useful insights regarding the spatiotemporal dynamics of NCad interactions in various biological processes, including ontogenesis and higher brain functions.

## Methods

**Plasmid construction**. All expression plasmids were subcloned into a pCAGGS1 vector[43]. To generate NCad-V, the PCR product encoding mVenus, flanked by a BamHI and NheI site, was inserted at amino acid position 311 of mouse NCad with a C-terminal HA tag. For the construction of NCad-GA, NCad-B1, and NCad-B3, we amplified ddGFP-A, ddFP-B1, and ddFP-B3, flanked by a BamHI and NheI site, from the synthesized gene fragments (Integrated DNA Technologies). In order to construct NCad-V$_{prox}$, we inserted mVenus, flanked by NheI sites, at amino acid position 714 of mouse NCad by overlapping PCR. NCad-GA$_{prox}$, NCad-B1$_{prox}$, and NCad-B3$_{prox}$ were generated by inserting ddGFP-A, ddFP-B1, and ddFP-B3, flanked by NheI sites, into NCad-V$_{prox}$ that the mVenus was excised by NheI, respectively. To distinguish cells expressing NCad-GA and NCad-B in the co-culture, marker fluorescent proteins mCherry and miRFP670 were tagged with a nuclear localization signal (NLS) and co-expressed bicistronically via a self-cleavable P2A peptide[44]. To generate NCad-GA with P2A-mCherry-NLS, NCad-B1 with P2A-miRFP670-NLS, and NCad-B3 with P2A-miRFP670-NLS, the PCR products encoding P2A-mCherry-NLS and P2A-miRFP670-NLS were inserted between an AgeI site and a NotI site. The AgeI site is located between the HA tag and the stop codon of NCad-GA, NCad-B1, and NCad-B3. The NotI site is derived from pCAGGS1. mCherry-NLS and EBFP2-NLS (Fig. 2) were constructed by subcloning the PCR products into the pCAGGS1 vector. In order to generate mutant NCad, we introduced four mutations by site-directed mutagenesis in combination with overlapping PCR. To establish stable cell lines, retrovirus plasmids were subcloned into pCX4puro and pCX4bsr vectors[45]. EBFP2-NLS and mCherry-NLS were subcloned into the pCX4bsr vector. Fluorescent protein-inserted NCad constructs were subcloned into the pCX4puro vector. For the construction of FRET-based NCad interaction indicators, C-Ncad and V-Ncad, we inserted Cerulean and Venus, flanked by transposon-derived extra amino acids, at amino acid position 311 of mouse NCad, by overlapping PCR according to a previous study design[18]. paavCAG-post-mGRASP-2A-dTomato (Addgene plasmid #34912) and paavCAG-pre-mGRASP-mCerulean (Addgene plasmid #34910) were gifted by Jinhyn Kim[24]. To construct post-mGRASP-P2A-mCherry-NLS and pre-mGRASP-P2A-EBFP2-NLS, post-mGRASP and pre-mGRASP fragments were amplified from paavCAG-post-mGRASP-2A-dTomato and paavCAG-pre-mGRASP-mCerulean and inserted into EcoRI/AgeI-digested NCad-GA-P2A-mCherry-NLS and NCad-B1-P2A-EBFP2-NLS plasmids, respectively.

**Animals**. Animal experimentation was performed according to the Institutional Guidelines on Animal Experimentation at Keio University. Pregnant ICR mice were purchased from Japan SLC (Japan) and housed under a 12 h light/12 h dark cycle in a temperature-controlled room. The day of vaginal plug detection was considered as embryonic day 0 (E0).

**Cell culture, plasmid transfection, and retroviral infection**. HEK293T cells (RIKEN BRC), L cells (ATCC), and COS7 cells (RIKEN BRC) were maintained in Dulbecco's modified Eagle's medium (DMEM, Sigma-Aldrich) supplemented with 10% (v/v) fetal bovine serum (FBS, Biowest) at 37 °C in humidified air containing 5% CO$_2$. K562 (RIKEN BRC) cells were maintained in Iscove's modified Dulbecco's medium (IMDM, Thermo Fisher Scientific) supplemented with 10 % (v/v) FBS. An ecotropic murine leukemia virus-packaging cell line, Platinum-E (PLAT-E)[46,47], was provided by Dr. Hideki Shibata (Nagoya University) and maintained in DMEM supplemented with 10% (v/v) FBS, 1 μg/mL of puromycin (InvivoGen), and 10 μg/mL of blasticidin (InvivoGen). HEK293T cells were transfected using polyethylenimine MAX (Cosmo Bio). COS7 cells were transfected using Lipofectamine LTX with Plus Reagent (Thermo Fisher Scientific). K562 cells were electroporated using an electroporator (NEPA21: NEPAGENE). L cells stably expressing NCad$^{WT}$-V and NCad$^{mut}$-V were established as follows: PLAT-E cells were transfected with pCX4puro-NCad$^{WT}$-V or pCX4puro-NCad$^{mut}$-V by FuGENE6 (Roche Diagnostics) according to the manufacture's protocol. After 48 h,

the cleared culture media were supplemented with 8 μg/mL of polybrene (Nakalai Tesque) and used for infection. More than 24 h after infection, L cells were selected with 5 μg/mL of puromycin. To establish L cells respectively expressing wild-type and mutant NCad-GA/mCherry-NLS, NCad-B1/EBFP2-NLS, and NCad-B3/EBFP2-NLS, we initially established L cells stably expressing mCherry-NLS and EBFP2-NLS by using PLAT-E cells transfected with pCX4bsr-mCherry-NLS and pCX4bsr-EBFP2-NLS, respectively. After establishing the cell lines, we introduced NCad-GA, NCad-B1, and NCad-B3 by using PLAT-E cells transfected with pCX4puro-NCad-GA, pCX4puro-NCad-B1, and pCX4puro-NCad-B3, respectively. Consequently, they were selected with 5 μg/mL of puromycin and 10 μg/mL of blasticidin.

For primary culture of cortical neurons with NCad-GA or NCad-B1, plasmids encoding NCad-GA-P2A-mCherry or NCad-B1-P2A-EBFP2 was introduced into the mouse embryonic cortical cells by in utero electroporation as previously described[48]. Three days later, electroporated areas in the cortices were dissected and dissociated with papain (Nakalai Tesque). mCherry or BFP positive cells were sorted from the dissociated cells by fluorescence activated cell sorting (FACS) (MoFlo XDP, Beckman). After sorting, an equal number of cells were mixed and plated onto a glass bottom dish (MatTek) coated with poly-D-lysine (Sigma-Aldrich) and cultured in Neurobasal Medium (Thermo Fisher Scientific) with B-27 supplement (Thermo Fisher Scientific) and L-glutamine (Sigma-Aldrich). For primary culture of neurons with C-Ncad or V-Ncad, cortices from embryonic day 15.5 ICR mice were dissected and dissociated with papain (Nakalai Tesque). Introduction of plasmids encoding C-Ncad or V-NCad to cortical cells was performed as previously described[48], with slight modifications. Briefly, in vitro electroporation was conducted using a NEPA21 electroporator (Nepagene) with the following conditions: $1 \times 10^7$ dissociated cells, 60 μg of plasmid, porting pulse (275 V, 0.5 ms pulse length, 50 ms interval, twice, 10% decay rate), and transfer pulse (20 V, 50 ms pulse length, 50 ms interval, 5 times, 40% decay rate). The cortical cells transfected with C-Ncad and those with V-Ncad were mixed in equal measure and plated onto a glass bottom dish (MatTek) coated with poly-D-lysine (Sigma-Aldrich) and cultured in Neurobasal Medium (Thermo Fisher Scientific) with B-27 supplement (Thermo Fisher Scientific) and L-glutamine (Sigma-Aldrich).

**Cell imaging**. For the co-culture experiment (Figs. 1b 5d, f and Supplementary Fig. 4c), HEK293T cells were individually transfected with expression plasmids and cultured for 24 h. The cells were then washed with PBS containing 1 mM EDTA and mixed in phenol red-free DMEM/F12 (Thermo Fisher Scientific) supplemented with 10% FBS. Cells were co-cultured in a 15 mL conical tube under slow rotation at room temperature for 8 h and seeded on a glass-bottomed dish coated with 0.1% (w/v) PEI (P3143, Sigma-Aldrich). Fluorescence and DIC images were acquired using an Olympus FV-1000 laser scanning confocal microscope with an IX81 microscope equipped with a ×40 1.35 numerical aperture (NA) oil-immersion objective lens (UApo/340 40×/1.35) or a ×60 1.35 NA oil-immersion objective lens (UPLSAPO60XO) (Olympus). Excitation wavelengths for EBFP2, INCIDER, Venus, mCherry, and miRFP670 were 405, 488, 488, 543, and 633 nm, respectively. In subsequent cell imaging experiments, cells transiently or stably expressing indicator proteins were seeded on glass-bottomed dishes coated with Cellmatrix Type IC (Nitta gelatin). After 24 to 48 h, the cells were observed using a confocal microscope.

For FRET ratio imaging, Cerulean was excited with a 405-nm laser. Donor and acceptor emissions were collected at 460–500 nm and 515–615 nm, respectively. An intensity display mode (IMD) image was created using MetaMorph software (Molecular Devices).

For neuronal cells at 2 DIV, cultured dishes were mounted in a 40% $O_2$ incubator chamber fitted onto a TCS SP8 laser scanning confocal microscope (Leica Microsystems). Images were acquired using the confocal microscope with a PL APO 40×/1.10 NA water-immersion or a PL APO CS2 63×/1.40 NA oil-immersion objective lens (Leica Microsystems). Excitation wavelengths for EBFP2, INCIDER, and mCherry were 405, 488, and 561 nm, respectively.

Two-photon imaging of L cell spheroids was performed as previously described[49]. Briefly, each separately maintained L cell line was dissociated to single cells and seeded in low-cell-adhesion 96-well plates (Sumitomo Bakelite) with the indicated combinations. To generate spheroids, we seeded 2000 cells for each cell line (total 4000 cells). Spheroids were cultured in DMEM supplemented with 10% (v/v) FBS. After 3 d, spheroids were placed on glass-bottomed dishes and immobilized with drops of undiluted Matrigel (BD Biosciences) for imaging. Two-photon fluorescent images were acquired using a custom-made inverted microscope (Olympus FV-1000/MPE combined with a $CO_2/O_2$ incubator) equipped with a ×25, 1.05 NA water-immersion objective lens XLPLN25XWMP2 (Olympus) and a Ti:sapphire laser Mai-Tai DeepSee eHP (Spectra-Physics). Excitation wavelengths for INCIDER and mCherry were 930 and 1,040 nm, respectively. We were unable to detect the EBFP2 signal due to the available microscopic optical setting.

Fluorescence images were analyzed using ImageJ[50]. Merged images were synthesized from fluorescence images without modification and DIC images by ImageJ through "Merge Channels" function.

**Western blot analysis**. Cells were lysed in lysis buffer (50 mM Tris-HCl, pH 7.6, 150 mM NaCl, and 1% Triton X-100) supplemented with Protease Inhibitor Cocktail for purification of Histidine-tagged proteins (Sigma-Aldrich) and incubated on ice for 30 min. After centrifugation at 18,000 g for 10 min at 4 °C, the cleared cell lysates were subjected to SDS-PAGE, followed by western blot using a rabbit polyclonal anti-human NCad antibody (×2000 dilution, Cat# M142, Takara Bio) and a mouse monoclonal anti-GAPDH antibody (×1000 dilution, clone 6C5, sc-32233, Santa Cruz Biotechnology).

**Surface staining**. Cells grown on 0.01% (w/v) poly-L-lysin (P8920, Sigma-Aldrich)-coated coverslips were fixed with 4% (w/v) paraformaldehyde in PBS (FUJIFILM) for 15 min at room temperature. The fixed cells were washed with PBS three times. After blocking with PBS containing 1% (w/v) BSA for 30 min at room temperature, the cells were labeled with a rabbit polyclonal anti-GFP antibody (×250 dilution, Code No. 598, MBL) for 1 h at room temperature. The cells were then washed with PBS containing 1% (w/v) BSA and incubated with an Alexa Fluor633-conjugated goat antibody against rabbit IgG (×1000 dilution, A21070, Thermo Fisher Scientific) for 1 h at room temperature. After washing with PBS containing 1% (w/v) BSA three times and rinsing with PBS, the coverslips were mounted on microscopic slides with Prolong Glass Antifade Mountant (Thermo Fisher Scientific).

**Cell aggregation assay**. Cell aggregation assay using K562 cells was performed as described previously[51]. Transfected K562 cells were rotated at 30 rpm overnight at 37 °C in humidified air containing 5% $CO_2$. Before imaging, the cells were transferred into glass-bottomed dishes coated with 0.1% (w/v) polyethylenimine (Sigma-Aldrich) for 1 h at 37 °C under humidified air containing 5% $CO_2$.

**Statistics and reproducibility**. Statistical analyses were performed using Graph-Pad Prism 9 (GraphPad Software, Inc.). Sample sizes are listed for each experiment. Reproducibility was confirmed by at least two independent experiments.

**Reporting summary**. Further information on research design is available in the Nature Research Reporting Summary linked to this article.

## Data availability

The data and materials supporting this research are available from the authors on reasonable request. Plasmid constructs encoding NCad-V (Addgene ID, 191665), NCad-GA (Addgene ID, 191666), NCad-B1 (Addgene ID, 191667), and NCad-B3 (Addgene ID, 191668) are available through Addgene. The source data underlying Figs. 1c, 2c, 3b, d, 4b, c, e, f, 5c, e, g, 6d, S2a, S2b, S3b, S5b, and S5d are provided as Supplementary Data. Uncropped images of western blots in Fig. 3a are provided in Supplementary Fig. 9.

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

## Acknowledgements

The authors are grateful to Hideki Shibata (Nagoya University) for providing retroviral vectors and packaging cells. We also thank Collaborative Research Resources, Keio University School of Medicine, for technical support of FACS. This work was supported by the MEXT Grant-in-Aid for Scientific Research on Innovative Areas "Interplay of developmental clock and extracellular environment in brain formation" (No. JP16H06487 to T.M., JP16H06485 to M.E., and JP16H06482 to K.N.), "Singularity biology" (No. JP18H05410) to T.N., JSPS Grant-in-Aid for Scientific Research (S) (No. JP20H05688) to K.N., JSPS Grant-in-Aid for Scientific Research (C) (No. JP18K06842) to K.H., JST PRESTO Program (No. JPMJPR2045) to T.K., JST CREST Program (No. JPMJCR20E4) to T.M. Takeda Science Foundation to K.N., Keio Gijuku Fukuzawa Memorial Fund for the Advancement of Education and Research to K.N., and Keio Gijuku Academic Development Funds to K.N.

## Author contributions

T.K. planned and performed experiments, analyzed data, and wrote the paper. Y.S. and K.H. performed experiments. M.E., K.N., and T.N. advised on the experiments and data analysis. T.M. designed the research plan and wrote the paper.

## Competing interests

The authors declare no competing interests.
