## [Peer Review File · Communications Biology]

Reviewers' comments:

Reviewer #1 (Remarks to the Author):

The authors report and begin to characterize a dimerization-dependent fluorescent protein-based optical tool to detect intercellular Ncadherin interactions.

The incompleteness of the manuscript distracts one from seeing any potential advantage of INCIDERS over the existing Ncad FRET reporter. There is missing information regarding sample size and number of independent experiments throughout the manuscript and most importantly, the authors do not provide quantification for the Western blot analysis and most of the imaging experiments. Without quantification and statistical analysis, the author's conclusions cannot be justified.

Major concerns:

1. There is no quantification of the experiments/images in Fig. 1b, 3b, 4a, 4b, 6a, 6b, 7c, and 7d.
2. In Figure 2, the authors report that INCIDER-1 has a higher fluorescence signal than INCIDER-3, which they attribute to the higher affinity between ddGFP-A and ddFP-B1. From their experiments in Figure 4, the authors conclude that the fluorescence signal is dependent on the Ncad interaction, and not based on INCIDER heterodimer formation. The result from Figure 2 suggests otherwise, yet this is completely ignored.
3. Sample sizes are missing in Figures 1, 3, 4, 6 and 7. Are the sample n's the number of cells or the number of interaction sites? How many independent experiments were conducted for each type of experiment?
4. Were the fluorescence intensity and imaging exposure the same in the experiments where fluorescence signal is being directly compared?
5. Other than comparing the signal-to-noise ratio between INCIDERS and Ncad FRET, are there other ways to demonstrate the improved ability to detect intercellular Ncad interactions? It does not seem fair to compare signal-to-noise ratios calculated from different parameters (ratiometric FRET vs. fluorescence intensity).

Minor concerns:

1. How the data were normalized in Fig 5b is not explained.

Reviewer #2 (Remarks to the Author):

In this study, Kanadome et al. developed and validated dimerization-dependent indicators of NCad interactions called INCIDERS. The authors adequately showed that co-culturing cells individually expressing the INCIDER components allows detection of intercellular NCad interactions. The authors confirmed that cis interactions between INCIDER components within a cell are negligible, and the fluorescence signal is primarily due to trans interactions between NCad molecules on adjacent cells. In evaluating performance of the indicators, the authors showed that INCIDERS have a high signal-to-background ratio compared to the previously developed FRET-based NCad interaction indicator, can detect changes in NCad interactions over time, and can be used in primary neuronal cells and in spheroids. Through a series of thorough experiments, the authors showed the validity and use of their novel ddGFP-based NCad indicators. The INCIDERS can have many potential uses to advance our understanding of NCad interactions in many different contexts. However, there are some key points that should be addressed before publication.

General comments:

The study lacks a method of quantification for INCIDER performance. Especially in figures 3b and 4b, some way of quantifying the interactions is needed. Perhaps the percent of interfaces between mCherry and BFP expressing cells that have INCIDER signal can be quantified for the different conditions. Alternatively, more representative images could be included in a supplementary figure. In addition, the number of cells analyzed should be reported for all experiments.

Specific Comments:

1. In the explanation for figure 1, the authors mention intracellular puncta. Perhaps providing an explanation of what they are will be beneficial for readers.
2. Figure 1 and 2 may be combined, and figure 1b, 2c, and 2d could be moved to a supplementary figure.
3. The loading control of the western blot in figure 4a is uneven. There should be repeats and quantification of the western blot. Also, the supplementary figure 2 showing the western blot is mislabeled.
4. In figure 4b, puncta structures are also seen, and there seems to be less membrane localization of the mutant NCad. Would loss of membrane localization contribute to loss of INCIDER signal in the NCad mutant expressing cell line? Giving an explanation or acknowledgement of this would provide clarity.
5. In figure 4, the authors show that INCIDER signal is dependent on NCad interactions, not self-interaction of INCIDER components. However, they have not addressed if dimerization is leading to or influencing contact site formation. At steady state, does dimerization encourage contact site formation? Comparing the percentage of contact site formation between cells expressing INCIDER components and cells expressing the FRET sensor components could clarify this. Comparing kinetics of the FRET sensor and INCIDER response to EGTA treatment can also be useful in this sense.
6. In figure 7a, only two neurons and one contact site are shown. Were more cells visualized? There should be inclusion of more images in the supplemental or a way of quantification. N number should also be reported.
7. For the spheroids, why did the authors not include BFP as a marker for cells expressing one of the INSIDER components? The cell-cell contact sites with NCad interactions are not clear, and many cells that are not expressing mCherry and are not near cells expressing mCherry have fluorescence. This experiment should be repeated with cells expressing BFP alongside one of the INCIDER components (design similar to the previous experiments). At a minimum, an explanation of why a marker was not included should be included, as well as arrows showing where NCad interactions are detected in the cells of the spheroids.

Reviewer #3 (Remarks to the Author):

In this study, Kanadome et al. developed INCIDERS as intensimetric indicators for visualizing N-cadherins interactions utilizing dimerization-dependent fluorescent proteins (ddFPs). Given ddFPs' properties that require no additional time for chromophore maturation, and mono/dimerization are reversible, the authors present that the INCIDERS enable real-time monitoring of cell-cell interactions with high SNR, compared to previous approaches such as FRET-, BiFC- and split GFP-based techniques. The authors applied the INCIDERS in dissociated neurons and spheroids. While authors' efforts to generate the ddFP-based indicators for cell-cell interaction are meaningful, my main concern, however, is inadequate technical and scientific novelty. The ddFP concept, constructs (GA, B1, B3), as well as design strategy for detecting NCads interactions have been previously published, as the authors stated in the manuscript. Therefore, I strongly suggest that further scientific applications need to be added in order to demonstrate the advantages of INCIDERS over the other previous strategies.

More specific concerns

The K_d of homodimerization of NCad is 25 μM , and the K_d values for GA-B1 and GA-B3 are 3 μM and 40 μM , respectively. How are the authors sure that the INCIDERS detect the cell-cell interaction, but not its self-dimerization? It would be interesting to make and test the B construct with a lower affinity. The authors tested two insertion sites, in the EC2 domain and membrane-proximal site. I am wondering whether other candidate sites can be tested.

Fig. 4a: The expression levels and patterns look not similar to me.

Fig.4b: Current combinations are Ncad wild types or Ncad mutants. How about the combinations of one mutant and one wild type?

Fig. 5: Reversibility is an essential advantage of the INCIDERS compared to BiFC and split GFP. Control experiments with these previous approaches will be helpful.

Also, I wonder whether further washout of EGTA can increase the fluorescence of INCIDERS again.

Fig. 6: Interestingly, the distribution of N-Cad is in the whole-cell when visualized by CFP or YFP, while the cis-interaction of NCad-GA and NCad-B was not that obvious in Fig. 3.

Also, I think it is not appropriate to compare the signal-to-noise ratios between FRET and ddFP technologies, as one is ratiometric and the other is intensimetric methods.

Fig. 7. Image quality is relatively poor. Replace other representative neurons and their neurites (morphology of mCherry-expressing neuron seems award, for instance).

Response to reviewers

We would like to thank the reviewers for their valuable comments and suggestions. In response to the reviewer's comments, we have provided point-by-point responses below.

Reviewer #1 (Remarks to the Author):

The authors report and begin to characterize a dimerization-dependent fluorescent protein-based optical tool to detect intercellular N-cadherin interactions.

The incompleteness of the manuscript distracts one from seeing any potential advantage of INCIDERS over the existing Ncad FRET reporter. There is missing information regarding sample size and number of independent experiments throughout the manuscript and most importantly, the authors do not provide quantification for the Western blot analysis and most of the imaging experiments. Without quantification and statistical analysis, the author's conclusions cannot be justified.

We deeply apologize for not providing full details of quantification, statistical analysis, and information regarding sample size and number of independent experiments. We have addressed these issues.

Major concerns:

1. There is no quantification of the experiments/images in Fig. 1b, 3b, 4a, 4b, 6a, 6b, 7c, and 7d.

We apologize for the absence of appropriate quantification and statistics. Since Fig. 1b (Supplementary Fig. 1 in current version) is merely to confirm whether co-expressed INCIDER components show fluorescence or not, we showed some representative images only and moved the content to Supplementary Fig. 1. In addition, since Fig. 7c (Supplementary Fig. 8 in current version) shows the formation of spheroids visualized in Fig. 7d (Fig. 6c in current version) as bright field images, we showed some representative images only and moved it to Supplementary Fig. 8. We quantified the images shown in Fig. 3b (Fig. 2b in current version), Fig. 4a, b (Fig. 3a, c in current version), Fig. 6a, b (Fig. 5a, b in current version), and Fig. 7d (Fig. 6c in current version) and included them in Fig. 2c, Fig. 3b, d, Fig. 5c, and Fig. 6d.

2. In Figure 2, the authors report that INCIDER-1 has a higher fluorescence signal than INCIDER-3, which they attribute to the higher affinity between ddGFP-A and ddFP-B1. From their experiments in Figure 4, the authors conclude that the fluorescence signal is dependent on the Ncad interaction, and not based on INCIDER heterodimer formation. The result from Figure 2 suggests otherwise, yet this is completely ignored.

Thank you for your critical comments. As you pointed out, our conclusion for Figure 4 needs further explanation to maintain consistency with the conclusion for Figure 2. Although Figure 4 indicates the indispensability of the NCad interaction in INCIDERS to cause fluorescence signals, it does not mean the signal strength is completely independent of the influence of heterodimerization of ddFPs. Interaction of NCad EC1-EC2 domains of INCIDERS would be necessary for appropriate ddFPs positioning to form a heterodimer for fluorescence. Once such configuration is set, the interaction of INCIDERS might be driven by the affinity of ddFPs. That is supported by the 1.5 times higher intensity of INCIDER-1 than INCIDER-3, reflecting a higher affinity of the ddFPs (ddGFP-A and ddFP-B1) contained in INCIDER-1 (Fig. 1c; Fig. 2b of the old version). Meanwhile, the difference in affinity did not contribute to a significant difference in the kinetics of the interaction (Fig. 3b-d; Fig. 4b, c of the old version) and the size of cell aggregate (Supplementary Fig. 3b). To reflect the above, we have added sentences in the main text as follows:

(page 17 Lines 281–290, in **Discussion** section)

“Here, we developed ddFP-based indicators for NCad interactions by inserting ddFPs in NCad. We showed that INCIDER signals were dependent on NCad interactions, not the self-interaction of ddGFP-A and ddFP-B (Fig. 3). However, INCIDER should generate the signal via the ddGFP-A/ddFP-B heterodimerization. This notion raises an interpretation that NCad interacting property of INCIDER leads to the formation of a ddGFP-A/ddFP-B heterodimer. Therefore, this ddGFP-A/ddFP-B heterodimerization possibly affects cell–cell interactions. Cell aggregation assay using K562 cells showed that INCIDERS induced cell aggregation to the same extent as Ncad-FRET, indicating that the heterodimerization of ddGFP-A/ddFP-B does not significantly induce cell–cell interactions (Supplementary Fig. 3).”

3. Sample sizes are missing in Figures 1, 3, 4, 6 and 7. Are the sample n's the number of

cells or the number of interaction sites? How many independent experiments were conducted for each type of experiment?

We apologize for not providing the sample numbers and the number of independent experiments. We have included them in each figure legend.

4. Were the fluorescence intensity and imaging exposure the same in the experiments where fluorescence signal is being directly compared?

Thank you for your comment. We also appreciate its importance for the direct comparison; to address this, we acquired images using a microscope under the same condition for that purpose.

5. Other than comparing the signal-to-noise ratio between INCIDERS and Ncad FRET, are there other ways to demonstrate the improved ability to detect intercellular Ncad interactions? It does not seem fair to compare signal-to-noise ratios calculated from different parameters (ratiometric FRET vs. fluorescence intensity).

We agree that it is not fair to compare the signal contrast calculated from different parameters to ensure scientific rigor. However, we included this comparison to explain the practical usefulness of INCIDERS against Ncad-FRET. Higher signal contrast of intensimetric indicators than that of FRET-based ratiometric indicators is generally known. It is typically shown for the well-engineered and investigated Ca^{2+} indicators. Therefore, intensimetric indicators have been used to readily detect signal changes in biological phenomena of interest even if the determination of absolute values is difficult. To show that this also occurs in the case of the imaging of Ncad interaction we showed the superiority of the INCIDERS for signal contrast since this was the best option. In support of the practical superiority of INCIDERS, we also include the data showing that the FRET signal of Ncad-FRET was not detected well in the neurons (Supplementary Fig. 7); in contrast, INCIDER-1 could detect NCad interaction across neuronal processes (Fig. 6a).

The usage of the term “signal-to-background ratios” is another aspect we reconsidered. For the comparison, we intended to compare the signal contrast between cell adhesion sites and non-cell adhesion sites. In that sense “signal-to-background

ratio” was not appropriate, since “background”, which generally defines a signal contrast between cells with and without expression of the indicator or, between the area with the cells expressing indicator cells and without cells, was not related to the calculation. Therefore, we corrected the term “signal-to-noise ratio” to “signal contrast” to avoid ambiguity (page 13-14, lines 231-243, **Comparison between INCIDERS and conventional indicators for cell–cell interactions** in the **Results** section).

Minor concerns:

1. *How the data were normalized in Fig 5b is not explained.*

We apologize for not explaining data normalization. We included the explanation in the figure legend (Fig. 4).

Reviewer #2 (Remarks to the Author):

In this study, Kanadome et al. developed and validated dimerization-dependent indicators of NCad interactions called INCIDERS. The authors adequately showed that co-culturing cells individually expressing the INCIDER components allows detection of intercellular NCad interactions. The authors confirmed that cis interactions between INCIDER components within a cell are negligible, and the fluorescence signal is primarily due to trans interactions between NCad molecules on adjacent cells. In evaluating performance of the indicators, the authors showed that INCIDERS have a high signal-to-background ratio compared to the previously developed FRET-based NCad interaction indicator, can detect changes in NCad interactions over time, and can be used in primary neuronal cells and in spheroids. Through a series of thorough experiments, the authors showed the validity and use of their novel ddGFP-based NCad indicators. The INCIDERS can have many potential uses to advance our understanding of NCad interactions in many different contexts. However, there are some key points that should be addressed before publication.

Thank you for your full understanding of our manuscript and comments to improve our manuscript.

General comments:

The study lacks a method of quantification for INCIDER performance. Especially in

figures 3b and 4b, some way of quantifying the interactions is needed. Perhaps the percent of interfaces between mCherry and BFP expressing cells that have INCIDER signal can be quantified for the different conditions. Alternatively, more representative images could be included in a supplementary figure. In addition, the number of cells analyzed should be reported for all experiments.

We apologize for not describing the method of quantification of INCIDER performance. We added quantification for the images shown in Fig. 3b and 4b (Fig. 2b and 3c in current version) in Fig. 2c and 3d, respectively and reported the sample numbers in each figure legend.

Specific Comments:

1. In the explanation for figure 1, the authors mention intracellular puncta. Perhaps providing an explanation of what they are will be beneficial for readers.

Thank you for your suggestion. We provided the explanation in the main text as follows:

(page 6-7, lines 105-108, **Design of ddFP-based NCad interaction indicators (INCIDERS)** in the **Results** section).

“While some artificial fluorescent puncta, which are considered autofluorescence were observed in the cells individually expressing components of the INCIDER (NCad-GA, NCad-B1, or NCad-B3)”

2. Figure 1 and 2 may be combined, and figure 1b, 2c, and 2d could be moved to a supplementary figure.

Thank you for your advice. We reorganized the figure as you suggested.

3. The loading control of the western blot in figure 4a is uneven. There should be repeats and quantification of the western blot. Also, the supplementary figure 2 showing the western blot is mislabeled.

We apologize for incompleteness of the western blot analysis. We repeated western blot analysis and quantified the data (Fig. 3a, b). We also revised the label of the uncropped

western blot figure (Supplementary Fig. 9).

4. In figure 4b, puncta structures are also seen, and there seems to be less membrane localization of the mutant NCad. Would loss of membrane localization contribute to loss of INCIDER signal in the NCad mutant expressing cell line? Giving an explanation or acknowledgement of this would provide clarity.

Thank you for your critical comments. A difference in membrane localization between wild-type and mutant NCad is very important. To address the concern, we performed surface staining of NCad^{WT}-V and NCad^{mut}-V stable cell lines and compared their membrane localization (Supplementary Fig. 5a). We found that NCad^{mut}-V localized at the plasma membrane with approximately 60% efficiency of NCad^{WT}-V (Supplementary Fig. 5b). This result implies that poorer membrane localization can partially contribute to loss of INCIDER signal in the NCad mutant expressing cell line as you pointed out. We included the results in Supplementary Fig. 5 and added sentences in the main text as follows:

(page 11-12 Lines 195–205, **Confirmation of signal specificity for NCad interactions** in the **Results** section)

“It is possible that a difference in membrane localization efficiency between NCad^{WT} and NCad^{mut} contributed to a difference in ddGFP signals. To address this possibility, we performed surface staining of NCad^{WT}-V-expressing cells and NCad^{mut}-V-expressing cells and estimated the efficiency by calculating the ratio of immunoreactive signals using anti-GFP antibody to Venus signals. We found that NCad^{mut}-V localized at the plasma membrane with approximately 60% efficiency of that of NCad^{WT}-V (Supplementary Fig. 4a, b). This result implies that poorer ddGFP signals from NCad^{mut}-GA and NCad^{mut}-B were partially contributed to by poorer membrane localization. Since a difference in membrane localization was not high compared with ddGFP signals, we concluded that INCIDERS give fluorescence dependent on the NCad-mediated cell–cell interaction.”

5. In figure 4, the authors show that INCIDER signal is dependent on NCad interactions, not self-interaction of INCIDER components. However, they have not addressed if dimerization is leading to or influencing contact site formation. At steady state, does

dimerization encourage contact site formation? Comparing the percentage of contact site formation between cells expressing INCIDER components and cells expressing the FRET sensor components could clarify this. Comparing kinetics of the FRET sensor and INCIDER response to EGTA treatment can also be useful in this sense.

Thank you for your critical comments. We agree that it is an important point. Although we attempted to follow your suggestion to address the concerns, unfortunately, we could not identify a good idea for proper identification of cell contact sites in the images. Instead, we examined the influence of INCIDERS or Ncad-FRET on the cell-cell contact site formation by cell aggregation assay using K562 cells. We found that cell aggregation induced by the expression of INCIDERS or Ncad-FRET was the same (Supplementary Fig. 3). Therefore, we concluded that the self-interaction of ddFPs does not significantly induce cell contact formation.

Following your suggestion, we also compared decay kinetics of INCIDERS and Ncad-FRET by EGTA treatment. The time scale of signal decay of INCIDERS was in the order of minutes-to-hour (Fig. 4a-c), while that of Ncad-FRET has been reported to be in the order of seconds (reference 18). These results mean that heterodimerization of ddFP can influence the dissociation of cell-cell contacts. While INCIDERS showed a relatively slow decay kinetics compared to Ncad-FRET, they are reversible, unlike BiFC-based indicators. Please evaluate the reversibility of INCIDERS compared with the lost reversibility of BiFC-based indicators.

We included this discussion point in the main text as follows:

(page 17 Lines 282–297, in **Discussion** section)

“We showed that INCIDER signals were dependent on NCad interactions, not the self-interaction of ddGFP-A and ddFP-B (Fig. 3). However, INCIDER should generate the signal via the ddGFP-A/ddFP-B heterodimerization. This notion raises an interpretation that NCad interacting property of INCIDER leads to the formation of a ddGFP-A/ddFP-B heterodimer; therefore, this ddGFP-A/ddFP-B heterodimerization possibly affects cell–cell interactions. Cell aggregation assay using K562 cells showed that INCIDERS induced cell aggregation to the same extent as Ncad-FRET, indicating that the heterodimerization of ddGFP-A/ddFP-B does not significantly induce cell–cell interactions (Supplementary Fig. 3). However, EGTA treatment experiments showed that time scale of fluorescence decay of INCIDERS upon EGTA treatment was in the order of minute-to-hour (Fig. 4a-c) although that of Ncad-FRET has been reported to be

in the order of seconds¹⁸. This means that the heterodimerization of ddGFP-A/ddFP-B possibly affects the dissociation of NCad interactions. While INCIDERS are reversible unlike BiFC-based indicators (Fig. 6f, g), it is noticeable that the relatively slow decay is a limitation of INCIDERS. The development of ddFP-B mutants with lower affinity to ddGFP-A may solve this limitation.”

6. In figure 7a, only two neurons and one contact site are shown. Were more cells visualized? There should be inclusion of more images in the supplemental or a way of quantification. N number should also be reported.

Thank you for your comments. We included additional images in Supplementary Fig. 6 and reported N number in the figure legend of Fig. 6a.

7. For the spheroids, why did the authors not include BFP as a marker for cells expressing one of the INSIDER components? The cell-cell contact sites with NCad interactions are not clear, and many cells that are not expressing mCherry and are not near cells expressing mCherry have fluorescence. This experiment should be repeated with cells expressing BFP alongside one of the INCIDER components (design similar to the previous experiments). At a minimum, an explanation of why a marker was not included should be included, as well as arrows showing where NCad interactions are detected in the cells of the spheroids.

We apologize for not explaining the reason why EBFP2 was not shown. We were unable to detect the EBFP2 signal in microscopic optical setting available to the authors of the experiment. We included the reason in the main text as follows:

(page 27 Lines 464–465, **Cell imaging** in the **Method** section)

“We were unable to detect the EBFP2 signal due to the available microscopic optical setting.”

As you suggested, we repeated the same experiments and obtained clearer images (Fig. 6c). Since arrows made it difficult to see the data, we did not add them in magnified panels. We also added the quantification (Fig. 6d).

Reviewer #3 (Remarks to the Author):

In this study, Kanadome et al. developed INCIDERS as intensimetric indicators for visualizing N-cadherins interactions utilizing dimerization-dependent fluorescent proteins (ddFPs). Given ddFPs' properties that require no additional time for chromophore maturation, and mono/dimerization are reversible, the authors present that the INCIDERS enable real-time monitoring of cell-cell interactions with high SNR, compared to previous approaches such as FRET-, BiFC- and split GFP-based techniques. The authors applied the INCIDERS in dissociated neurons and spheroids. While authors' efforts to generate the ddFP-based indicators for cell-cell interaction are meaningful, my main concern, however, is inadequate technical and scientific novelty. The ddFP concept, constructs (GA, B1, B3), as well as design strategy for detecting NCads interactions have been previously published, as the authors stated in the manuscript. Therefore, I strongly suggest that further scientific applications need to be added in order to demonstrate the advantages of INCIDERS over the other previous strategies.

Thank you for your full understanding of the manuscript concepts and for providing suggestions to improve our manuscript. As you mentioned, it is true that INCIDERS lack technical and scientific novelty. The novelty of our study lies in demonstrating further scientific applications. We compared the reversibility between INCIDERS and a BiFC-based indicator according to your suggestion to address of your specific concerns. We also showed the practicality of INCIDER-1 in neurons compared to Ncad-FRET.

More specific concerns

The K_a of homodimerization of NCad is 25 μM , and the K_a values for GA-B1 and GA-B3 are 3 μM and 40 μM , respectively. How are the authors sure that the INCIDERS detect the cell-cell interaction, but not its self-dimerization? It would be interesting to make and test the B construct with a lower affinity.

Thank you for your comments. We confirmed that INCIDERS detect the cell-cell interaction via the NCad interacting property, but not its self-dimerization, by using mutant INCIDERS in which *cis* and *trans* interaction sites of NCad are destroyed (Fig. 3). We revised a sentence for better clarity of the concern you pointed out before showing the results as follows: (page 10, lines 177-179).

(page 10 Lines 177–179, **Confirmation of signal specificity for NCad interactions** in the **Results** section)

“Since ddGFP-A and ddFP-B have an affinity for heterodimer formation, INCIDERS might generate signals only via the self-interaction of ddGFP-A and ddFP-B regardless of NCad interactions.”

We agree that it would be interesting to make and test a ddFP-B construct with a lower affinity. We expect the development of a new ddFP-B construct in the future.

The authors tested two insertion sites, in the EC2 domain and membrane-proximal site. I am wondering whether other candidate sites can be tested.

Thank you for your comments. We have tested only two sites as you mentioned. It might be worth testing other candidate sites in the future.

Fig. 4a: The expression levels and patterns look not similar to me.

We deeply apologize for not providing a full description of the western blot analysis. We repeated the experiments and quantified the data (Fig. 3a, b).

Fig.4b: Current combinations are Ncad wild types or Ncad mutants. How about the combinations of one mutant and one wild type?

Thank you for your suggestion. We tried the WT/mut combination as you suggested. We found that the WT/mut pair showed ddGFP signal, which was lower than that of the WT/WT pair. We included the results in Supplementary Fig 5c, d and mentioned them in the main text as follows: Results section (page 12, lines 206-211).

(page 12 Lines 206–211, **Confirmation of signal specificity for NCad interactions** in the **Results** section)

“We further performed the co-culture in the combination of NCad^{WT}/NCad^{mut}. Both NCad^{WT}-GA/NCad^{mut}-B1 pair and NCad^{mut}-GA/NCad^{WT}-B1 pair showed ddGFP

signals that were lower than the NCad^{WT}-GA/NCad^{WT}-B1 pair (Supplementary Fig. 5c, d). This result raises the possibility that NCad^{WT} weakly interacts with NCad^{mut}. However, further analysis is required to reach a conclusion as we did not examine the interaction between NCad^{WT} and NCad^{mut}.”

Fig. 5: Reversibility is an essential advantage of the INCIDERS compared to BiFC and split GFP. Control experiments with these previous approaches will be helpful.

Thank you for your constructive suggestion. We included the comparison experiments between INCIDERS and mGRASP, a BiFC-based indicator, in Fig. 5d-g and mentioned them in the main text as follows:

(pages 14-15 Lines 244–255, **Comparison between INCIDERS and conventional indicators for cell–cell interactions** in the **Results** section)

“Next, we compared the performance between INCIDERS and a BiFC-based indicator for cell–cell interactions (mGRASP²⁴). mGRASP showed green fluorescence at the cell adhesion sites similar to INCIDERS and its signal was more than twice as high as that of INCIDERS (Fig. 5d, e). BiFC-based indicators for cell–cell interactions are potentially unable to monitor the dissociation of cell–cell interactions. To evaluate the relative dissociation property, we examined the effect of EGTA treatment on pre-formed cell–cell interactions. We first co-cultured HEK293T cells individually expressing INCIDERS or mGRASP components for 6 h and then further cultured them in the presence or absence of 10 mM EGTA for 2 h. The INCIDER signals were diminished by incubation with EGTA while mGRASP showed signals regardless of EGTA treatment (Fig. 5f, g). This result shows that INCIDERS can monitor the dissociation of cell–cell interactions unlike BiFC-based indicators.”

Also, I wonder whether further washout of EGTA can increase the fluorescence of INCIDERS again.

Thank you for your constructive comment. We examined the effect of further washout of EGTA on the fluorescence of INCIDERS and found that the fluorescence of INCIDERS was recovered. We included the results in Fig. 4d-f and mentioned them in the main text as follows: Results section (page 13, lines 223-227).

(page 13 Lines 223–227, **Time-lapse imaging of reversible NCad interaction** in the **Results** section)

“We further examined whether INCIDERS can monitor a re-association of NCad. INCIDER signals that declined with EGTA treatment were increased with the washout of EGTA again (Fig. 4d-f). These results indicate that INCIDERS show the appropriate availability for monitoring the transition between association and dissociation of NCad interactions.”

Fig. 6: Interestingly, the distribution of N-Cad is in the whole-cell when visualized by CFP or YFP, while the cis-interaction of NCad-GA and NCad-B was not that obvious in Fig. 3.

Thank you for your comment. Co-expressed NCad-GA and NCad-B should localize at the whole cells similarly to C-Ncad and V-Ncad (Fig. 5a). However, ddGFP signals were predominantly observed at the cell adhesion sites (Fig. 2b in current version, Fig. 3b in previous version). Our interpretation is that co-expressed NCad-GA and NCad-B preferentially visualize the *trans*-interaction, but not the *cis*-interaction of NCad.

Also, I think it is not appropriate to compare the signal-to-noise ratios between FRET and ddFP technologies, as one is ratiometric and the other is intensimetric methods.

Thank you for your comment. Firstly, we would like to revise the expression of “signal-to-back-ground ratio”. We noticed that the expression of signal-to-background ratios is not appropriate since what we want show is defined as signal contrast between the cell adhesion sites and non-cell adhesion sites. Signal-to-background ratio can be generally considered as contrast between indicator-expressing cells and non-expressing cells or between indicator-expressing cells and regions without cells. Using the expression of signal-to-background ratio can mislead readers. Therefore, we changed the “signal-to-back ground ratio” to “signal contrast” for clarity.

It may be true that comparison of the signal contrast is not appropriate. However, it is well accepted that intensimetric indicators show high signal contrast or high signal change compared to ratiometric indicators based on FRET. Thereby, intensimetric indicators have been used to readily detect signal change in biological phenomena of

interest. To examine the practicality of this approach, we applied Ncad-FRET to neurons; however, FRET signal was not detected well (Supplementary Fig. 7). By contrast, INCIDER-1 could detect NCad interaction across processes (Fig. 6a). Please evaluate the high signal contrast of INCIDERS and its practicality as an advantage against Ncad-FRET.

Fig. 7. Image quality is relatively poor. Replace other representative neurons and their neurites (morphology of mCherry-expressing neuron seems awkward, for instance).

We apologize for providing poor quality images. We repeated the experiments and replaced the images (Fig. 6a, Supplementary Fig.6).

Reviewers' comments:

Reviewer #1 (Remarks to the Author):

In this study, Kanadome et al. develop and validate a robust method called INCIDERS to detect intercellular N-cadherin interactions based on a dimerization-dependent fluorescent protein (ddFP) strategy. The authors clearly address the advantages of INCIDERS over other methods to label membrane contacts between cells by demonstrating a ~70-fold higher signal contrast compared to the existing FRET-based NCad interaction indicator and the ability to visualize reversible intercellular NCad interactions not possible with the biomolecular fluorescence complementation (BiFC)-based indicator mGRASP. The authors have made significant revisions to the manuscript by providing the necessary data quantification and statistical analysis to substantiate their conclusions and by including additional experiments that further demonstrate the superior utility of INCIDERS over other cell-cell interaction detection methods. INCIDERS is a useful tool to assess NCad interactions in various cellular contexts; however, there are a few minor points that should be addressed before the manuscript is suitable for publication.

1. Regarding Supplementary Figure 1: It seems that the fluorescence intensity in the merged images (bottom panel) is higher than the single channel images in the upper panel. The intensity settings in the fluorescence images (top panel) and the fluorescence+DIC merged images (bottom panel) should be kept exactly the same or it should be acknowledged in the figure legend that they are different.
2. Regarding Figure 1: it should be stated somewhere in either the main text or the figure legend that the NCad-GA expressing cells were co-expressing mCherry, and that the NCad-B-expressing cells were co-expressing mRFP670. Without this explanation, the statements "mCherry-NLS and mRFP670-NLS are hallmarks of NCad-GA-expressing cells and NCad-B-expressing cells, respectively" (in the figure legend) and "cells expressing NCad-GA and NCad-B were distinguished by the nuclear localized fluorescent marker proteins mCherry and mRFP670, respectively" (in the main text), are confusing.
3. Will the constructs designed in this study be made available on Addgene?
4. Regarding Figure 3: Are both bands (corresponding to processed and unprocessed Ncad) in the Western blot used in the quantification in 3b? The authors state the expression levels of unprocessed and processed NCad varied between the cell lines to some extent, but didn't quantify this directly. Since NCad processing/cleavage occurs in the ER/Golgi, and it does look like there's more Golgi accumulation of the mutant NCad-Venus compared to WT NCad-Venus (Supplementary Figure 5a), I wonder if quantification of processed vs. unprocessed NCad levels across cell lines might help explain why the mutant NCad-Venus is less efficient at trafficking to the surface. A blot with more uniform loading would help visualize whether there's relatively equal amounts of unprocessed (adhesively inactive) and processed NCad.
5. Regarding Figure 4: How long after EGTA washout were the cells imaged? References should be added about the reversibility and irreversibility of ddFPs and BiFC-based indicators, respectively.
6. Regarding Supplementary Figure 7: The neurons in the representative images appear less healthy than those shown in Figure 6a and Supplementary Figure 6. Was the FRET efficiency calculated in the experiments shown in Supplementary Figure 7, and are they comparable to the FRET efficiencies reported in the original publication (as a control measure)?

Reviewer #2 (Remarks to the Author):

The reviewers' comments have been appropriately addressed.

Reviewer #3 (Remarks to the Author):

The authors addressed my previous concerns and the manuscript has been improved with appropriate quantification data.

The authors additionally conducted the requested experiments, in particular, the new data (in Fig. 4d-f and Fig. 5d-g) confirming the reversibility of its responses emphasize the advantage of the INCIDERS well.

I have only one minor suggestion as follows.

In the results section for new figure 5 (page 14, line 244), mGRASP is introduced as a BiFC-based indicator, but split-GFP would be more appropriate description for the technique of mGRASP. Because BiFC, in general, refers to the technique of reconstituting bimolecular fluorescent protein fragments consisting of 7 and 4 strands (e.g. GFP-N and GFP-C), respectively. In contrast, mGRASP is composed of fragments consisting of 1-10 strands and 11 strand, which is called split-GFP technique.

Response to reviewers

We would like to thank the reviewers for their valuable comments and suggestions. In response to the reviewer's comments, we have provided point-by-point responses below.

Reviewer #1 (Remarks to the Author):

In this study, Kanadome et al. develop and validate a robust method called INCIDERS to detect intercellular N-cadherin interactions based on a dimerization-dependent fluorescent protein (ddFP) strategy. The authors clearly address the advantages of INCIDERS over other methods to label membrane contacts between cells by demonstrating a ~70-fold higher signal contrast compared to the existing FRET-based NCad interaction indicator and the ability to visualize reversible intercellular NCad interactions not possible with the biomolecular fluorescence complementation (BiFC)-based indicator mGRASP. The authors have made significant revisions to the manuscript by providing the necessary data quantification and statistical analysis to substantiate their conclusions and by including additional experiments that further demonstrate the superior utility of INCIDERS over other cell-cell interaction detection methods. INCIDERS is a useful tool to assess NCad interactions in various cellular contexts; however, there are a few minor points that should be addressed before the manuscript is suitable for publication.

We are very grateful to reviewer #1 for the positive evaluation of our revised manuscript and for further comments helpful in improving our manuscript.

1. Regarding Supplementary Figure 1: It seems that the fluorescence intensity in the merged images (bottom panel) is higher than the single channel images in the upper panel. The intensity settings in the fluorescence images (top panel) and the fluorescence+DIC merged images (bottom panel) should be kept exactly the same or it should be acknowledged in the figure legend that they are different.

Thank you for your comment. Since we used exactly the same fluorescence images without modification for merging, the intensity of the fluorescence channel should be preserved in the merged images. However, as you pointed out, apparent fluorescence in the merged images (bottom panel) looks brighter than the single channel images (top panel), probably due to the influence of the merged DIC channel. For the preparation of

the merged images we processed by ImageJ as follows:

1. We opened Image>Color>Merge Channels.
2. We selected a green channel for a fluorescence image and a gray channel for a DIC image and also selected “Create composite” and clicked “OK”.
3. We went on Image>Color>Stack to RGB to obtain RGB images.

Following your suggestion, to clarify the way to generate the merged images we added the following sentence in the Methods section.

(page 27 Lines 467–469, **Cell imaging** in the **Methods** section)

“Merged images were synthesized from fluorescence images without modification and DIC images by ImageJ through ‘Merge Channels’ function.”

2. Regarding Figure 1: it should be stated somewhere in either the main text or the figure legend that the NCad-GA expressing cells were co-expressing mCherry, and that the NCad-B-expressing cells were co-expressing miRFP670. Without this explanation, the statements “mCherry-NLS and miRFP670-NLS are hallmarks of NCad-GA-expressing cells and NCad-B-expressing cells, respectively” (in the figure legend) and “cells expressing NCad-GA and NCad-B were distinguished by the nuclear localized fluorescent marker proteins mCherry and miRFP670, respectively” (in the main text), are confusing.

We apologize for our confusing description. We deleted the sentence “mCherry-NLS and miRFP670-NLS are hallmarks of NCad-GA-expressing cells and NCad-B-expressing cells, respectively” in the figure legend and added the new sentence as follows:

(page 40 Lines 675–676, in the **figure legend**)

“mCherry-NLS and miRFP670-NLS were bicistronically co-expressed with NCad-GA and NCad-B, respectively to distinguish cells.”

3. Will the constructs designed in this study be made available on Addgene?

Thank you for your comment. We will supply materials supporting this research on reasonable request as described in the **Data availability** section. We are also planning to deposit constructs in Addgene for open access if the paper is to be accepted.

4. Regarding Figure 3: Are both bands (corresponding to processed and unprocessed Ncad) in the Western blot used in the quantification in 3b?

We used both bands (processed and unprocessed NCad) in the western blot analysis for the quantification in Fig. 3b.

The authors state the expression levels of unprocessed and processed NCad varied between the cell lines to some extent, but didn't quantify this directly.

We apologize for misleading you. We just wanted to state that “expression levels of total NCad containing both processed and unprocessed forms” not “processing efficiency of NCad” varied among cell lines to some extent. We revised the sentence in the results section and added a new sentence in the figure legend.

(page 11 Lines 186–189, **Confirmation of signal specificity for NCad interactions in the Results** section)

“We confirmed, by western blot analysis using an anti-NCad antibody, that NCad was detected as two bands (upper and lower bands corresponding to unprocessed and processed NCad, respectively) and expression levels of total NCad varied between cell lines to some extent (Fig. 3a, b).”

(page 44 Lines 706–707, in the **figure legend**)

“Both upper and lower bands of NCad were used for the quantification.”

Since NCad processing/cleavage occurs in the ER/Golgi, and it does look like there's more Golgi accumulation of the mutant NCad-Venus compared to WT NCad-Venus (Supplementary Figure 5a), I wonder if quantification of processed vs. unprocessed NCad levels across cell lines might help explain why the mutant NCad-Venus is less efficient at trafficking to the surface. A blot with more uniform loading would help visualize whether there's relatively equal amounts of unprocessed (adhesively inactive) and processed NCad.

As you mentioned, it is one possibility that a difference in processing efficiency between mutant NCad and wild-type NCad contributes to the difference in their membrane

localization efficiency. As another possibility, we also suppose that mutant NCad which is unable to form a trans-interaction across cells is not trapped at the membrane and thus can easily undergo endocytosis, resulting in lesser membrane localization of mutant NCad. To address these possibilities, evaluation of processing efficiency based on western blot as you suggested and examining the cellular localization by cellular fractionation or indirect immunofluorescence would be helpful. However, revealing the reason for the difference in the membrane localization efficiency between mutant and wild-type NCad deviates from the main claim in this study. While it is interesting to study the basic property of the NCad molecule, investigation of the mutant, which lost the function of the NCad indicator, seems difficult to link with the improvement of the NCad indicator. We expect that this issue will be dealt with as a separate future effort.

5. Regarding Figure 4: How long after EGTA washout were the cells imaged?

Thank you for your comment. We acquired images after 3 h from EGTA washout. We described the experimental details in the figure legend as follows:

(page 47 Lines 736–739, in the **figure legend**)

“HEK293T cells individually expressing INCIDER-1 components (NCad-GA and NCad-B1) were observed at 1h after seeding. After 2 h from 10 mM EGTA treatment, cells were washed to remove EGTA and further cultured for 3 h under a microscope.”

References should be added about the reversibility and irreversibility of ddFPs and BiFC-based indicators, respectively.

We apologize for not providing appropriate references. We cited references about the reversibility of ddFPs in the introduction section as follows:

(page 5 Lines 77–80, in the **Introduction** section)

“Unlike split-GFP technique, ddFPs do not require a time delay after protein–protein interactions and are reversible, which allows for real-time monitoring of transitions between interactions^{31–35}.”

We also cited references about the irreversibility of BiFC (split-GFP, in current version)-based indicators in the introduction and results section as follows:

(page 4 Lines 65–66, in the **Introduction** section)

“the irreversibility of the process makes it impossible to monitor the dissociation of cell–cell interactions^{25,28,30}.”

(page 14 Lines 247–249, **Comparison between INCIDERS and conventional indicators for cell–cell interactions** in the **Results** section)

“Split-GFP-based indicators for cell–cell interactions are potentially unable to monitor the dissociation of cell–cell interactions^{25,28,30}.”

6. Regarding Supplementary Figure 7: The neurons in the representative images appear less healthy than those shown in Figure 6a and Supplementary Figure 6.

Thank you for your comment. As you mentioned, we noticed that the neurons in Supplementary Fig.7 appear less healthy. CFP and YFP signals were quite low so we unavoidably set the laser power high. Phototoxicity by intense light would induce neurons unhealthy.

Was the FRET efficiency calculated in the experiments shown in Supplementary Figure 7, and are they comparable to the FRET efficiencies reported in the original publication (as a control measure)?

We did not calculate the FRET efficiency in Supplementary Fig.7. In the original paper of the FRET-based NCad indicator (reference 18) FRET efficiency in neurons was not reported, since the indicator was not applied to the imaging of neurons. Furthermore, to our knowledge, there is no report in which FRET-based NCad indicator was applied in neurons. Therefore, we cannot directly compare FRET efficiency between our and other work as a control measure.

Reviewer #2 (Remarks to the Author):

The reviewer’s comments have been appropriately addressed.

We are very grateful to reviewer 2 for the positive evaluation of our manuscript.

Reviewer #3 (Remarks to the Author):

The authors addressed my previous concerns and the manuscript has been improved with appropriate quantification data.

The authors additionally conducted the requested experiments, in particular, the new data (in Fig. 4d-f and Fig. 5d-g) confirming the reversibility of its responses emphasize the advantage of the INCIDERS well.

We are very grateful to reviewer #3 for the positive evaluation of our revised manuscript and a useful comment.

I have only one minor suggestion as follows.

In the results section for new figure 5 (page 14, line 244), mGRASP is introduced as a BiFC-based indicator, but split-GFP would be more appropriate description for the technique of mGRASP. Because BiFC, in general, refers to the technique of reconstituting bimolecular fluorescent protein fragments consisting of 7 and 4 strands (e.g. GFP-N and GFP-C), respectively. In contrast, mGRASP is composed of fragments consisting of 1-10 strands and 11 strand, which is called split-GFP technique.

Thank you for your suggestion. We agree that split-GFP, not BiFC is appropriate description for the technique of mGRASP. Since mixed usage both “BiFC” and “split-GFP” brings confusing, we uniformed the description as “split-GFP” in whole text.

REVIEWERS' COMMENTS:

Reviewer #1 (Remarks to the Author):

The authors have appropriately addressed my previous concerns.